# Systems genetics in the rat HXB/BXH family identifies *Tti2* as a pleiotropic quantitative trait gene for adult hippocampal neurogenesis and serum glucose

**Anna N. Senko**[1,2], **Rupert W. Overall**[1,2], **Jan Silhavy**[3], **Petr Mlejnek**[3], **Hana Malínská**[4], **Martina Hüttl**[4], **Irena Marková**[4], **Klaus S. Fabel**[1,2], **Lu Lu**[5], **Ales Stuchlik**[3], **Robert W. Williams**[5], **Michal Pravenec**[3], **Gerd Kempermann**[1,2]*

**1** German Center for Neurodegenerative Diseases (DZNE) Dresden, Germany, **2** CRTD–Center for Regenerative Therapies Dresden, Technische Universität Dresden, Germany, **3** Institute of Physiology of the Czech Academy of Sciences, Prague, Czech Republic, **4** Institute for Clinical and Experimental Medicine, Prague, Czech Republic, **5** Department of Genetics, Genomics and Informatics, University of Tennessee Health Science Center, Memphis, Tennessee, United States of America

* gerd.kempermann@dzne.de, gerd.kempermann@tu-dresden.de

**Data Availability Statement:** Neurogenesis data was deposited in the HXB/BXH Published Phenotypes Database at the GeneNetwork (www.

## Abstract

Neurogenesis in the adult hippocampus contributes to learning and memory in the healthy brain but is dysregulated in metabolic and neurodegenerative diseases. The molecular relationships between neural stem cell activity, adult neurogenesis, and global metabolism are largely unknown. Here we applied unbiased systems genetics methods to quantify genetic covariation among adult neurogenesis and metabolic phenotypes in peripheral tissues of a genetically diverse family of rat strains, derived from a cross between the spontaneously hypertensive (SHR/Olalpcv) strain and Brown Norway (BN-*Lx*/Cub). The HXB/BXH family is a very well established model to dissect genetic variants that modulate metabolic and cardiovascular diseases and we have accumulated deep phenome and transcriptome data in a FAIR-compliant resource for systematic and integrative analyses. Here we measured rates of precursor cell proliferation, survival of new neurons, and gene expression in the hippocampus of the entire HXB/BXH family, including both parents. These data were combined with published metabolic phenotypes to detect a neurometabolic quantitative trait locus (QTL) for serum glucose and neuronal survival on Chromosome 16: 62.1–66.3 Mb. We subsequently fine-mapped the key phenotype to a locus that includes the Telo2-interacting protein 2 gene (*Tti2*)—a chaperone that modulates the activity and stability of PIKK kinases. To verify the hypothesis that differences in neurogenesis and glucose levels are caused by a polymorphism in *Tti2*, we generated a targeted frameshift mutation on the SHR/Olalpcv background. Heterozygous SHR-*Tti2*⁺/⁻ mutants had lower rates of hippocampal neurogenesis and hallmarks of dysglycemia compared to wild-type littermates. Our findings highlight *Tti2* as a causal genetic link between glucose metabolism and structural brain plasticity. In humans, more than 800 genomic variants are linked to *TTI2* expression, seven of which have associations to protein and blood stem cell factor concentrations, blood pressure and frontotemporal dementia.

genenetwork.org) under Record IDs: HRP_10193 (BrdU+ cells; http://genenetwork.org/show_trait?trait_id=10193&dataset=HXBBXHPublish), HRP_10194 (Ki67+ cells; http://genenetwork.org/show_trait?trait_id=10194&dataset=HXBBXHPublish), HRP_10195 (clusters of Ki67+ cells; http://genenetwork.org/show_trait?trait_id=10195&dataset=HXBBXHPublish). Hippocampal gene expression dataset was deposited at GeneNetwork under GN Accession GN231 (http://gn1.genenetwork.org/webqtl/main.py?FormID=sharinginfo&GN_AccessionId=231&InfoPageName=UT_HippRatEx_RMA_0709). RNA sequencing of SHR and SHR-Tti2+/- rats was deposited to Gene Ontology Omnibus under the accession number GSE160361 (https://www.ncbi.nlm.nih.gov/geo/query/acc.cgi?acc=GSE160361).

**Funding:** This work was supported by DFG (KE 615/9-1) (ANS, GK), Helmholtz-Gemeinschaft EnergI (ANS, GK). MP and HM were supported by the Ministry of Health of the Czech Republic under the conceptual development of research organizations program (Institute for Clinical and Experimental Medicine – IKEM, IN 00023001 (HM) and by the Academic Premium (Praemium academiae) (AP1502) (MP); AS was supported by Czech Science Foundation (GACR) project 20-00939S. RWW received support from the NIGMS Systems Genetics and Precision Medicine Project (R01 GM123489, 2017-2026) and NIDA NIDA Core Center of Excellence in Transcriptomics, Systems Genetics, and the Addictome (P30 DA044223, 2017-2022). The funders had no role in study design, data collection and analysis, decision to publish, or preparation of the manuscript.

**Competing interests:** The authors have declared that no competing interests exist.

## Author summary

Metabolic and neurological disorders are often comorbid, suggesting that biological pathways which orchestrate peripheral homeostasis and the integrity of the nervous system intersect. The genetic architecture behind these relationships is still poorly described, in part because molecular processes in the human brain are very difficult to study. We thus used a rodent genetic reference population to investigate links between adult hippocampal neurogenesis—a cellular plasticity mechanism important for learning flexibility—and metabolism. We measured adult neurogenesis in the family of 30 HXB/BXH rat recombinant inbred strains, who are characterised by stable differences in metabolism, behaviour, and gene expression levels.

Because DNA variants affecting distinct traits segregated into different members of the family, it was possible to determine which of the previously published phenotypes correlated to adult neurogenesis due to shared genomic sequence. We found that expression levels of *Tti2*—a part of a specialised protein chaperone complex regulating stability of PIKK kinases—were concomitantly influencing adult neurogenesis and serum glucose levels. In human populations hundreds of genomic variants regulate *TTI2* expression, potentially affecting brain function and glucose homeostasis.

## Introduction

Epidemiological studies link components of the metabolic syndrome—a complex disorder characterised by the coexistence of obesity, insulin resistance, dyslipidaemia, and hypertension—to cognitive impairment and dementia [1]. Defective brain function is often seen as a consequence of longstanding metabolic dysregulation. The full picture, however, is more complex. Several human phenome-wide association studies identified causal genetic loci that are shared between metabolic and neurological phenotypes [2–8], suggesting some degree of pleiotropy—a phenomenon whereby one gene variant affects multiple traits. Pleiotropy is widespread among model organisms [9–14] and humans [2,15–19] with an estimated median number of around six traits per locus [13]. Pleiotropic mutations can result in genetic covariance among phenotypes [10,12,20]. Cognition and metabolic homeostasis are both achieved through multiple elementary mechanisms at molecular, cellular, tissue, and inter-organ levels, some of which might be shared. To understand the biology underlying correlations between such complex functions, it is necessary to identify which exact processes are simultaneously affected by shared genetic variation.

Adult neurogenesis in the dentate gyrus (DG) of the hippocampus is required for cognitive flexibility of learning, efficient pattern separation and emotional processing in mammals [21], and thus embodies a functionally relevant and readily quantifiable parameter of hippocampal plasticity. In humans, according to the best available calculation, one third of dentate granule cells are born during adulthood [22]. The generation of new granule neurons in the DG is a complex multistep process, in the course of which neural precursor cell proliferation, as well as survival, maturation, and integration of newly born postmitotic cells are under the control of multiple genetic loci. Remarkably, cellular metabolism has been identified as a regulator of neural stem and progenitor cell maintenance, proliferation, and differentiation [23], yet the interactions of adult neurogenesis with systemic metabolism are far from being understood. Given the overall significance of metabolism for brain function in health and disease, there is a

high likelihood that direct causal links exist between structural brain plasticity and metabolic traits and states.

Natural genetic variation present in genetically diverse mouse populations contributes to up to ten-fold differences in the net production of new neurons [24,25]—values much larger than the induction achieved by environmental interventions within a single genetic background. This enormous genetic potential inherent in rodent strains can be utilised to dissect the molecular interaction networks not only underlying adult neurogenesis as such, but also its connections with homeostatic mechanisms in peripheral tissues.

We thus employed a family of 30 fully inbred recombinant HXB/BXH rat strains, which have been derived by reciprocal mating of a spontaneously hypertensive rat line (SHR/OlaIpcv, hereafter referred to as SHR) with the normotensive BN-*Lx*/Cub—a Brown Norway (BN) congenic rat with a mutation that causes a polydactyly-luxate phenotype [26,27]. Besides hypertension, SHR manifests other hallmarks of metabolic syndrome [28–30], as well as cognitive deficits [31,32], and brain morphological changes [33,34]. The HXB/BXH family was developed to map quantitative trait loci (QTL) for hypertension and morphological abnormalities associated with polydactyly-luxate syndrome [26,27], but they are of great utility for genetic mapping of a much wider spectrum of phenotypes. The HXB/BXH family have become the most thoroughly characterised rat reference population within the Rat Hybrid Diversity Panel [35]. Over 200 metabolic, endocrinological, behavioural and developmental phenotypes, along with gene expression profiles in peripheral tissues, are publicly accessible in a FAIR-compliant format at the GeneNetwork database [36,37]. The stable genetic background of these recombinant inbred lines enables truly systemic integration of phenotypic and omics data [35]. Accordingly, the HXB/BXH family have been used to clone genes associated with several disease quantitative trait loci (QTL) and discover gene regulatory networks relevant for human diseases [29,30,38–44]. Genome sequencing of both founder strains revealed 3.2 million single nucleotide variants, 425,924 small insertions and deletions, 907 copy number changes, and 1,094 large structural genetic variants segregating within the HXB/BXH family [45,46]. This level of genetic diversity allows for the separation of phenotypes modulated by different sets of gene variants. Correlations between phenotypes, in turn, suggest shared genetic causality [10,20,47].

In the present study, we quantified adult hippocampal neurogenesis in all existing HXB/BXH family members and parents. We aimed at finding out the extent of genetic correlations between neurogenesis and peripheral metabolism, and sought to identify common loci that cause these correlations. The data reported here were submitted into public databases as a part of the HXB/BXH phenome resource.

## Results

### Adult neurogenesis in HXB/BXH strains

We quantified adult hippocampal neurogenesis in the DG of young male rats from all 30 HXB/BXH family members, as well as the parental founder strains BN and SHR. Numbers of proliferating stem and progenitor cells were estimated using immunohistochemistry against the Ki67 antigen (Fig 1A and 1D). Dividing cells were also labelled with BrdU at 10 weeks of age and surviving progeny were quantified four weeks later (Fig 1B and 1E), after which time point new cells are likely to persist for very long periods of time [48,49]. Newborn cells were identified as neurons by double labelling with neuronal marker, NeuN (Rbfox3, Fig 1C). Among 3814 BrdU$^+$ cells in 38 individuals, we detected only a single BrdU$^+$/S100β$^+$ astrocyte. We conclude that either astrogliogenesis is negligible in the DG of young HXB/BXH rats, or S100β (the standard marker in mice in this case) is not expressed in newly-born astrocytes in

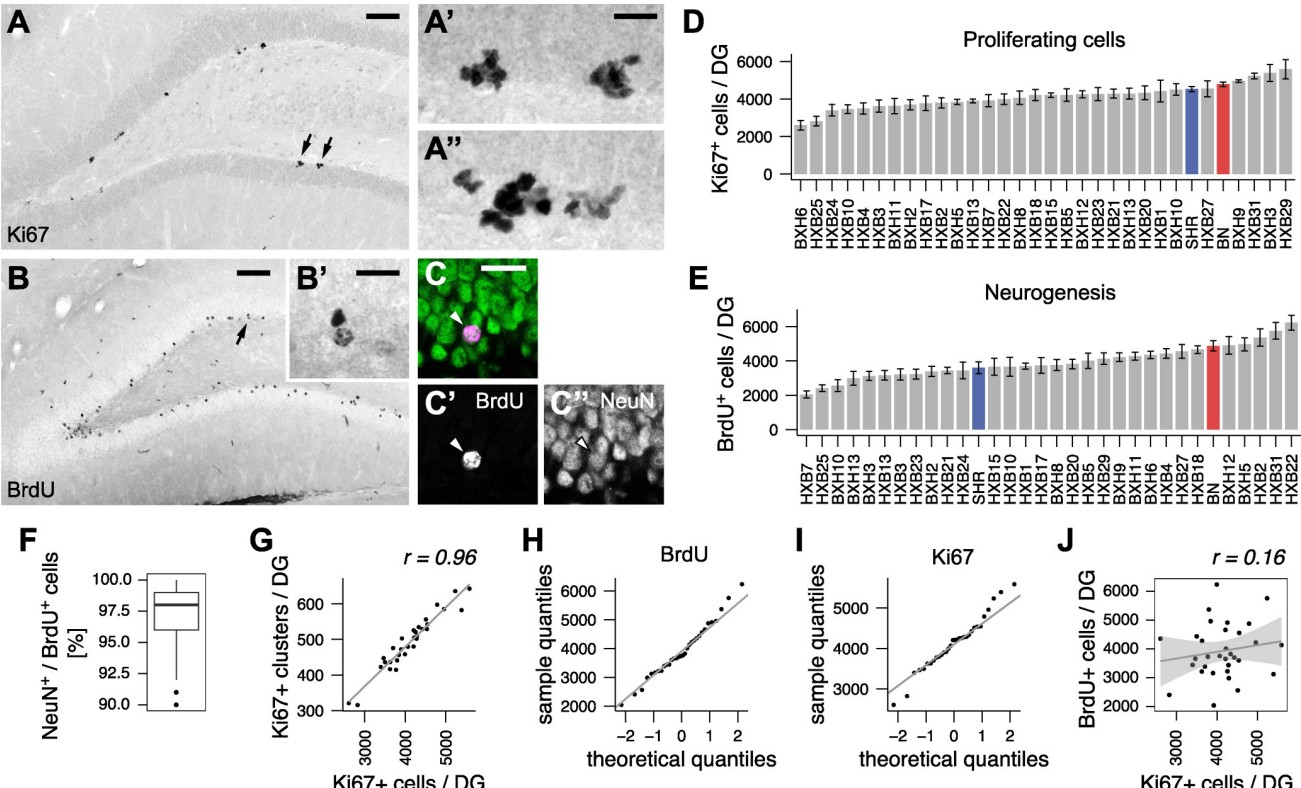

**Fig 1. Adult neurogenesis in the HXB/BXH family.** (A) Proliferating precursor cells were stained for Ki67, a marker of actively cycling cells. (A'-A") Proliferating cells occur in tightly packed clusters, which are likely to arise from single activated stem cells. (A') High power view of two clusters indicated by arrows in (A). (A") An example of a single large cluster of Ki67+ cells. (B) New cells in the DG were identified by DAB-immunostaining for BrdU 4 weeks after the last BrdU injection. (B') High power view of cells indicated by an arrow. (C) BrdU-positive cells (magenta, arrowhead) were identified as neurons by confocal microscopy after co-labelling with a neuronal marker, NeuN (green). (D-E) The number of proliferating (D) and surviving (E) cells in the DG of HXB/BXH, BN (red) and SHR (blue) strains. (F) Distribution of neuronal percentages among newborn cells across HXB/BXH family confirms that more than 90% of newborn cells are neurons. Box and whisker plot: centre line—median; upper and lower hinges—first and third quartiles; whiskers—highest and lowest values within 1.5 times the interquartile range outside hinges; dots—outlying data points. (G) Correlation between number of clusters and individual counts of proliferating cells. (H-I) Quantile-quantile plots of neurogenesis traits indicate normal distribution of strain means. (J) Rates of precursor cell proliferation are not predicting the number of surviving cells. *r*, Pearson's product-moment correlation coefficient. Scale bar, 100 and 20 μm in A and B, 20 μm in A', A", B" and C.

these lines. Notably, over 90% of BrdU-positive cells were neurons (Fig 1F). Hence, we used the numbers of surviving BrdU-positive cells as an approximation of net neurogenesis within the HXB/BXH family. The majority of dividing cells in the DG are transient amplifying progenitors, which remain tightly clustered together (Fig 1A–1A"). We quantified the number of such clusters of proliferating cells, assuming that each cluster arises from an activated stem cell. The number of clusters tightly correlated to counts of individual cells (Pearson's $r^2 = 0.92$; Fig 1G). Such high correlation suggested that the numbers of proliferating cells were determined by the numbers of activated stem cells rather than differences in lineage progression or cell cycle dynamics in these animals, and pointed to a stereotyped pattern of lineage progression.

Proliferation and survival differed between 2- and 3-fold across the strains (Fig 1D and 1E) and were normally distributed (Fig 1H and 1I; BrdU: $W = 0.983$, $p = 0.88$; Ki67: $W = 0.976$, $p = 0.67$; Shapiro-Wilk test), consistent with the polygenic regulation of adult neurogenesis. Transgressive segregation was observed for both traits, indicating that genes with positive effects on neurogenesis were distributed between the parental strains. The heritability was

estimated as 0.41 and 0.29 for survival and proliferation, respectively. Interestingly, proliferation levels did not predict the rates of survival, as these measures did not correlate with each other ($r^2$ = 0.026; Fig 1J). This finding implies that, under standard laboratory conditions, these two aspects of neurogenesis are influenced by largely separate sets of genes.

## Shared QTL for neurogenesis and serum glucose

The search for genetic associations between adult neurogenesis and physiological traits was performed in several steps. First, we identified QTL for both neurogenic traits (Fig 2A and 2B). We detected a suggestive QTL for net neurogenesis on chromosome 16 (LOD = 3.12, genome-wide corrected *p* value = 0.14), and a weak suggestive QTL for proliferation on chromosome 17 (LOD = 2.39, *p* = 0.52). Second, we identified phenotypes from the GeneNetwork database [37] that significantly correlated to both traits (Table 1). We then used these phenotypes as covariates in conditional QTL scans to screen for potential interactions at the genomic level. A substantial change in the LOD score after using another phenotype as a covariate indicates that genetic variation within a QTL may have pleiotropic effects on these two phenotypes [50]. Among pairs of correlating phenotypes, we detected only one such association: QTL mapping for net neurogenesis revealed a LOD drop below suggestive level to 0.43 after conditioning on serum glucose levels. Accordingly, an overlapping significant QTL for serum glucose was found on chromosome 16 (LOD = 5.13, *p* = 0.0065; Fig 2C). This locus explained 37% of the genetic variance in adult neurogenesis and 61% of the genetic variance in serum glucose. The SHR allele was associated with an average decrease in numbers of BrdU-positive cells by 584 and in serum glucose by 0.656 mmol/L.

All phenotypes were measured in male rats aged between 6 to 10 weeks. Details can be found in the GeneNetwork database (www.genenetwork.org). *r*, Pearson's product correlation coefficient; N, number of overlapping strains; ID, GeneNetwork HXB/BXH Published Phenotypes identifier.

Finally, to derive a confidence interval for the joint survival-serum glucose QTL, we combined the variance from both traits using their first principal component, here referred to as an eigenphenotype (Fig 2D). The eigenphenotype QTL (LOD = 4.62, *p* = 0.014) spanned 4.2 Mb from genomic position 62.1 to 66.3 Mb and contained 10 protein-coding and 4 non-coding RNA genes (S1 Table). The QTL covered 4983 single nucleotide variations and 1803 small insertions/deletions ($\leq$ 10 base pairs).

## *Tti2* is a candidate quantitative trait gene

Genetic correlations between phenotypes can result from variation in shared regulatory genes or from linkage disequilibrium. In linkage disequilibrium, distinct genes governing each phenotype co-segregate together in a limited population of recombinant strains due to their physical proximity in the genome. To distinguish these scenarios and prioritise candidate genes for each phenotype, we used transcriptional profiles in tissues relevant for adult neurogenesis and metabolic regulation to cross-correlate with phenotypes and genetic markers. To that end, we profiled gene expression data from hippocampi of HXB/BXH and parental strains using microarrays. We also used published gene expression data sets from the soleus muscle, liver, perirenal fat, kidney, adrenal gland, left ventricle and aorta. We identified 2893 genes whose expression correlated to net neurogenesis in the hippocampus or to serum glucose in at least one peripheral tissue with an absolute value of Pearson's *r* equal to or larger than 0.4. Among these genes, 3 were located within the common neurogenesis-glucose QTL interval on chromosome 16: Telo2-interacting protein 2 (*Tti2*), Neuregulin 1 (*Nrg1*), and *Mak16*. Serum glucose correlated to expression of *Nrg1* in the muscle (*r* = 0.47), and fat (*r* = 0.44), and to *Mak16*

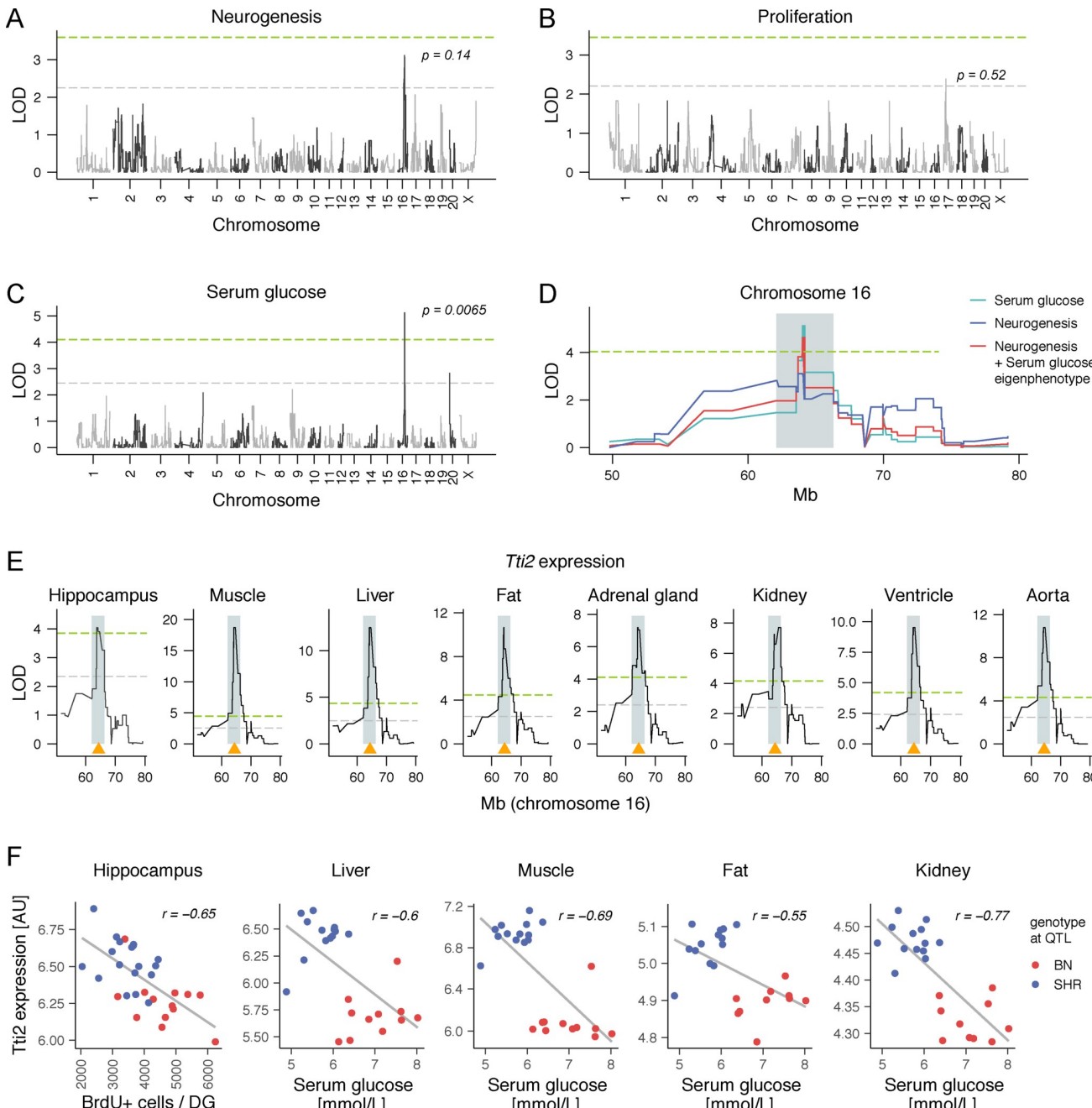

**Fig 2.** ***Tti2* is a candidate gene for common net neurogenesis and serum glucose quantitative trait locus (QTL).** (A-E) Whole-genome quantitative trait locus mapping for indicated traits. The genome-wide significant ($p < 0.05$) and suggestive ($p < 0.63$) thresholds of the logarithm of the odds (LOD) score (green and grey horizontal dashed lines, respectively) were calculated using permutations and corrected *p*-values are shown adjacent to the highest association in (A-C). Suggestive threshold corresponds to an average of one false positive per genome scan [51,52]. QTL for net adult neurogenesis and a positively correlating phenotype, serum glucose level, have an overlapping pattern on chromosome 16. (D) An eigenvector of the two phenotypes (an *eigenphenotype*–the first principal component) was used to calculate a common confidence interval for the shared QTL (shaded areas in D and E). (E) *Tti2* gene, whose genomic position is indicated by an orange triangle, has a local expression QTL within the eigenphenotype QTL confidence interval on chromosome 16 in all tested tissues. For clarity, in (D) and (E) only fragments of the chromosome 16 were plotted. (F) *Tti2* expression in the hippocampus or tissues relevant for regulation of glucose homeostasis, as indicated in the figure, is correlated to net neurogenesis and serum glucose. Colour specifies parental genotypes at the marker, which had the highest LOD score association with the eigenphenotype.

**Table 1. Published phenotypes correlating to adult neurogenesis in HXB/BXH family.**

| Phenotype | *r* | N | *p*-value | ID |
|---|---|---|---|---|
| **Net neurogenesis (BrdU$^+$ cells)** | | | | |
| Glucose concentrations | 0.57 | 24 | 0.0031 | 10003 |
| Serum triglyceride concentrations | 0.46 | 24 | 0.023 | 10014 |
| Liver triglycerides | 0.41 | 29 | 0.026 | 10119 |
| Relative kidney weight | -0.40 | 28 | 0.034 | 10025 |
| Serum chromogranin A levels | 0.37 | 30 | 0.043 | 10132 |
| **Proliferation (Ki67$^+$ cells)** | | | | |
| Adrenal phenyletanolamine-N-mythyltransferase | 0.58 | 30 | 0.0006 | 10151 |
| Adrenal dopamine | 0.52 | 30 | 0.0031 | 10106 |
| Adrenal epinephrine | 0.47 | 30 | 0.0081 | 10105 |
| Relative kidney weight | -0.45 | 28 | 0.015 | 10025 |
| Insulin stimulated lipogenesis in epididymal fat | -0.44 | 31 | 0.012 | 10148 |
| Adrenal chromogranin A levels | 0.43 | 30 | 0.016 | 10133 |
| Serum corticosterone levels after immobilization stress | -0.42 | 23 | 0.043 | 10064 |
| Serum triglyceride concentrations, fed high fructose diet for 2 weeks | 0.42 | 24 | 0.039 | 10016 |
| Adrenal norepinephrine | 0.40 | 30 | 0.029 | 10107 |
| Urine calcium | 0.39 | 28 | 0.041 | 10179 |
| Basal lipogenesis in epididymal fat | -0.38 | 31 | 0.033 | 10146 |

in the muscle (*r* = -0.59), aorta (-0.56), and adrenal gland (*r* = -0.53). *Tti2* was the only gene from the entire transcriptome whose expression correlated significantly to both traits across all data sets (Fig 2F). Finally, *Tti2* mRNA expression was linked to a multi-tissue cis-eQTL mapping within the neurogenesis-glucose QTL (Fig 2E). Other genes that had cis-eQTL within the joint QTL interval were *Mak16* in the muscle (LOD = 6.11), and *Saraf* in the kidney (LOD = 4.4) (S2 Table). The *Saraf* gene itself is located outside of the phenotype QTL, and its expression did not correlate to physiological phenotypes.

We thus carried out conditional mapping for neurogenesis and serum glucose using expression of *Tti2*, and *Mak16* as covariates (S2 Table). A substantial drop in the LOD score between unconditioned and conditioned scans strongly suggests that the QTL effect is mediated by the gene expression—consistent with the flow of causation from genes to phenotypes [50,53,54]. Using *Tti2* expression in the hippocampus as a covariate decreased the LOD score for adult neurogenesis by 2.34 (LOD = 0.78). To investigate the effect of *Tti2* expression in peripheral tissues on serum glucose mapping, we first summarised *Tti2* expression levels in these data sets as an eigengene (the first principal component). Using the *Tti2* eigengene as a covariate in QTL mapping decreased the LOD score for serum glucose by 2.76 (LOD = 2.37). The conditional LOD values for both phenotypes were below suggestive level. Using *Mak16* expression in the muscle as a covariate also decreased the LOD score for serum glucose by 2.31 (LOD = 2.82), but not for net neurogenesis (LOD = 3.16). The genomic locus of *Mak16* partly overlaps *Tti2* gene (S1 Table), so it is plausible they share some of the regulatory elements. Based on the multi-tissue cis-eQTL and correlation of *Tti2* expression to neurogenesis and serum glucose in all data sets, we concluded it is the strongest candidate gene responsible for observed variation in both phenotypes within the joint phenotype QTL. The *Tti2* SHR allele was associated with higher levels of *Tti2* mRNA, which we verified using quantitative RT-PCR in RNA isolated from the hippocampus, liver, muscle, kidney and pancreas (S3 Table).

The allelic variation underlying a QTL can either change the expression level of a gene, or the function of its product by altering its structure. We inspected the genes located within the

eigenphenotype QTL confidence interval for non-synonymous amino acid substitutions. Interestingly, only *Tti2* carried several missense mutations, including one at a highly conserved position, although none of the substitutions were predicted as damaging by Polymorphism Phenotyping 2 (Polyphen 2) [55] or Sorting Intolerant From Tolerant (SIFT) [56] (S4 Table). Together, these data support *Tti2* as a causal candidate gene for the combined serum glucose and net neurogenesis QTL.

## Reduced *Tti2* expression impairs adult neurogenesis and metabolic homeostasis

To evaluate whether expression of *Tti2* might be indeed causally linked to regulation of net adult neurogenesis and serum glucose, we derived heterozygous *Tti2* knockout rats on the SHR background. Using zinc finger nuclease, we introduced an 8-bp deletion in the first exon of *Tti2*. The deletion resulted in a frameshift mutation Glu82Gly, which generated 9 premature stop codons within the N-terminal portion of the protein, truncating Tti2 from 502 to 112 amino acids. Heterozygous male rats were compared to wild type littermates to assess consequences of the reduction of available Tti2 for adult hippocampal neurogenesis and metabolism.

Net adult neurogenesis decreased by 21% in the DG of three-month-old SHR-*Tti2*$^{+/-}$ compared to their wild-type SHR littermates (Fig 3A and Table 2). Concomitantly, heterozygous animals exhibited extensive alterations in metabolic parameters (Figs 3 and S1 and Tables 2 and 3). Heterozygous knock-out of *Tti2* resulted in lower plasma glucose and insulin levels (Fig 3B and 3C). SHR-*Tti2*$^{+/-}$ rats had also elevated plasma triglycerides (TG; Fig 3D) and non-esterified fatty acids (NEFA; Fig 3F) compared to SHR control. Changes in plasma lipid profile were accompanied by a decrease in liver TG and cholesterol content (Fig 3G and 3H). However, we did not observe an effect on total or HDL-bound fraction of plasma cholesterol (Figs 3E and S1H). In SHR-*Tti2*$^{+/-}$ animals we observed changes in body composition and organ sizes, including lipogenic tissues: they had smaller livers and epididymal fat deposits and increased perirenal fat weights (S1B–S1D Fig). On the other hand, knock-out of *Tti2* did not affect the weight of brown adipose tissue (BAT; S1E Fig). While the heterozygous rats were slightly larger compared to the control animals (S1A Fig), this difference was not statistically significant.

Table 2 reports means ± standard errors of the mean. *p* values were derived from Student's *t*-test or post-hoc Tukey test following two-way mixed effect model with an interaction between genotype and stimulation as a fixed factor. Asterisks denote significance at $p < 0.05$. Abbreviations: BAT, brown adipose tissue; CAT, catalase; HDL-C, high-density lipoprotein bound cholesterol; epid., epididymal; GPx, glutathione peroxidase; GR, glutathione reductase; GSH, glutathione; GSSG, oxidised glutathione; NEFA, non-esterified fatty acids; SOD, superoxide dismutase; TBARS, thiobarbituric acid reactive substances; TG, triglycerides. Raw data can be found in S1 Data.

Glucose and lipid metabolism were measured *ex vivo* in tissues isolated from SHR-*Tti2*$^{+/-}$ rats and wild type SHR littermates under basal and induced conditions (without or with 250 μU/ml insulin or 250 mg/ml adrenaline). Table 3 shows results of two-way linear mixed effect models with Insulin, Genotype and Insulin:Genotype interaction or Adrenaline, Genotype and Adrenaline:Genotype as fixed effects and individual intercepts as a random effect. Statistical significance was evaluated by likelihood ratio test. Adjusted *p* values for comparisons within each genotype were obtained from *post-hoc* Tukey test for the interaction. Means and SEM for each group and the comparison between genotypes are shown in Table 2. Asterisks denote significance at $p < 0.05$; d.f., degrees of freedom. Raw data can be found in S1 Data.

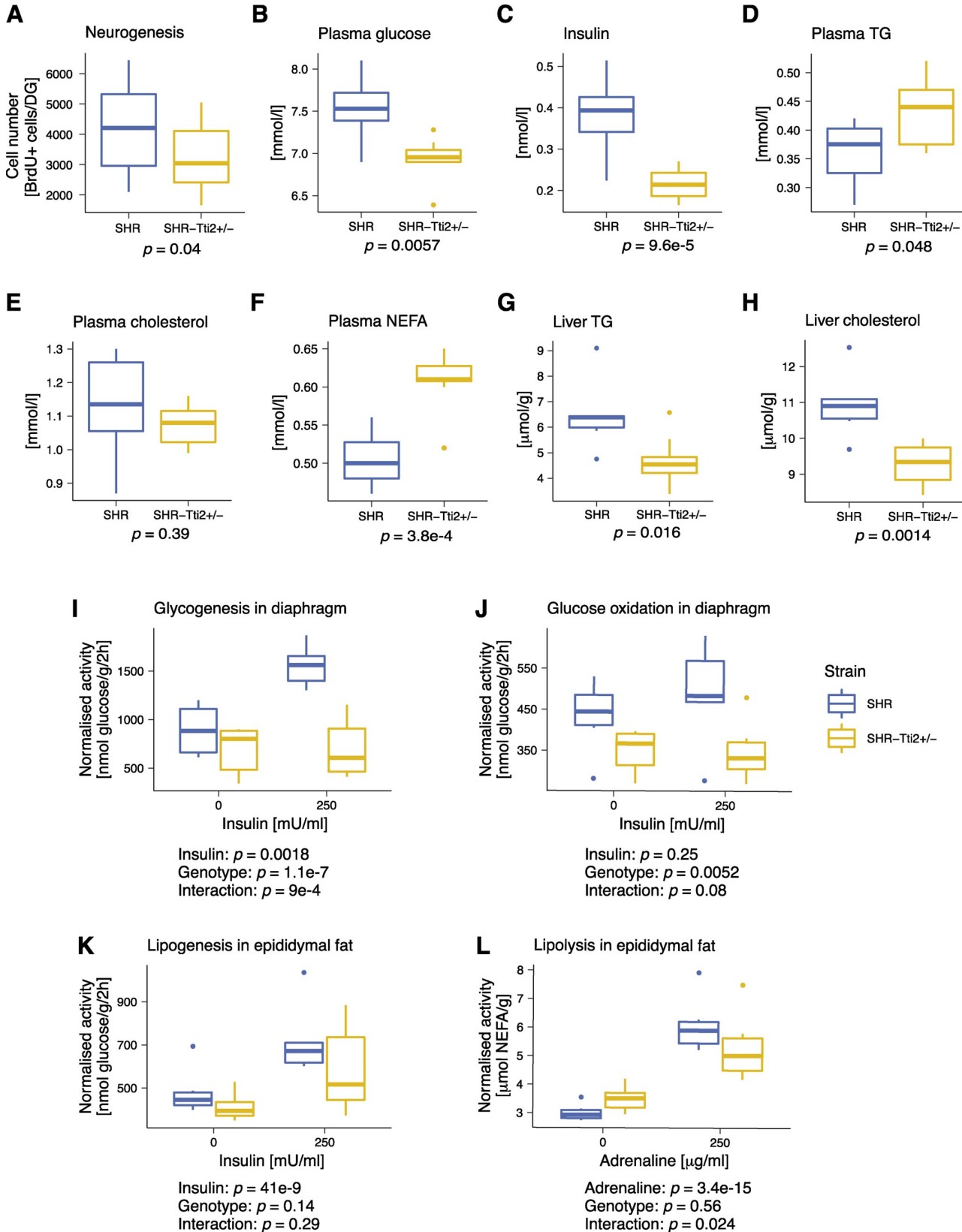

**Fig 3. Knock-down of *Tti2* leads to decreased hippocampal neurogenesis and impaired glucose homeostasis.** (A-H) Three-month-old heterozygous SHR-*Tti2*$^{+/-}$ rats and wild type SHR-*Tti2*$^{+/+}$ littermates (denoted as SHR) were assessed for the phenotypes indicated in the figure. (I-K) Glucose and lipid metabolism were measured *ex vivo* in diaphragm or epididymal adipose tissue in absence (basal conditions) or presence of 250 μU/mL insulin (stimulated condition) in the incubation media. (L) Basal and adrenaline-stimulated lipolysis were measured in the epididymal adipose tissue in absence or presence of 250 μg/ml adrenaline. Number of animals: (A) 18 rats of each genotype; (B-L) 8 SHR-*Tti2*$^{+/-}$, 6 SHR. *p* values were derived from Student's *t*-test (A-H) or linear mixed effect model (I-L). Full details of statistical analysis are in Tables 2 and 3. Raw data can be found in S1 Data. Abbreviations: NEFA, non-esterified free fatty acids; TG, triglycerides.

Next, we investigated rates of glucose and lipid metabolism *ex vivo* in the skeletal muscle and adipose tissues from SHR-*Tti2*$^{+/-}$ and SHR control rats under basal conditions and upon stimulation. In the diaphragm, knock-out of *Tti2* abolished stimulatory effect of insulin on both glucose incorporation into glycogen, and glucose oxidation (Fig 3I and 3J and Tables 2 and 3). However, the basal rates of glucose oxidation and glycogenesis in heterozygous rats were not significantly different from wildtype littermates. Similarly, basal glucose oxidation in BAT was not different between genotypes (S1M Fig). In contrast to glucose utilisation in the skeletal muscle, incorporation of glucose into lipids in epididymal adipose tissue was significantly stimulated by insulin in both genotypes (Fig 3K) and we did not detect differences between heterozygous and control rats in basal nor stimulated condition. *De novo* lipogenesis also did not differ between SHR-*Tti2*$^{+/-}$ and control rats in BAT (S1N Fig). Furthermore, both SHR-*Tti2*$^{+/-}$ and SHR animals upregulated lipolysis in the presence of adrenaline (Fig 3L). However, a significant interaction between genotype and adrenaline stimulation suggested a different response to stimulation depending on the expression level of *Tti2*.

Metabolic deregulation and dyslipidaemia are often associated with elevated oxidative stress. We thus examined the hallmarks of the hepatic oxidative status. Indeed, significantly upregulated conjugated dienes in livers from heterozygous rats compared to control littermates (Fig 4A) suggested increased oxidative stress. Increased oxidation was indicated also by decreased ratio of reduced to oxidised glutathione (GSH/GSSG; Fig 4G), mostly due to increased concentrations of oxidised glutathione (GSSG; Fig 4H). In contrast, the content of thiobarbituric acid reactive substances (TBARS) showed a decreasing trend (*p* = 0.095; Fig 4B). The changes in lipid peroxidation were accompanied by decreased activity of the antioxidant enzymes, superoxide dismutase (SOD; Fig 4C) and glutathione peroxidase (GPx; Fig 4E). Glutathione reductase (GR; Fig 4F) and catalase (CAT, Fig 4D) were not significantly different between SHR-*Tti2*$^{+/-}$ and control littermates.

Together, these results indicate that lowering expression of functional full-length *Tti2* affected adult neurogenesis and metabolism, in particular glucose and insulin homeostasis, in agreement with *Tti2* being a causal gene underlying the joint neurogenesis and serum glucose QTL.

## Gene expression profiles indicate glucose intolerance in SHR-*Tti2*$^{+/-}$ rats

The Tti2 protein, together with its binding partners, telomere maintenance 2 (Telo2) and Tti1, form a chaperone complex that assists folding and tertiary assembly of functional phosphatidylinositol 3-kinase-related kinases (PIKK): mammalian target of rapamycin (mTOR), ataxia telangiectasia mutated (ATM), ataxia telangiectasia and Rad3 related (ATR), suppressor of morphogenesis in genitalia (SMG1), transformation/transcription domain-associated protein (TRRAP) and DNA-dependent protein kinase catalytic subunit (DNA-PKcs) [57]. To get an insight into the molecular consequences of downregulation of *Tti2* expression, we performed transcriptional profiling of the hippocampus and three tissues essential for metabolic regulation in SHR-Tti2$^{+/-}$ rats and their wild-type littermates. We detected 326 differentially expressed transcripts in the liver, 52 in the soleus muscle, 41 in the perirenal fat, and 10 in the hippocampus. In addition, the analysis confirmed reduction of *Tti2* mRNA in the heterozygous rats (S5

**Table 2. Statistical comparison of neurogenesis and metabolic traits between SHR-*Tti2*+/- rats and SHR wild type littermates.**

| Phenotype | Unit | SHR | SHR-*Tti2*+/- | *p* value |
|---|---|---|---|---|
| **Neurogenesis** | BrdU+ cells/DG | 4143 ± 79 | 3266 ± 55 | 0.04 * |
| **Body and organ weights** | | | | |
| Body weight | g | 296.47 ± 2.62 | 308.41 ± 1.37 | 0.12 |
| Relative weight of epid. fat | g/100 g bwt | 0.69 ± 0.003 | 0.63 ± 0.005 | 0.0049 * |
| Relative weight of perirenal fat | g/100 g bwt | 0.52 ± 0.01 | 0.58 ± 0.01 | 0.022 * |
| Relative weight of BAT | g/100 g bwt | 0.09 ± 0.002 | 0.08 ± 0.002 | 0.71 |
| Relative weight of liver | g/100 g bwt | 3.48 ± 0.01 | 3.3 ± 0.01 | 0.0061 * |
| Relative weight of heart | g/100 g bwt | 0.36 ± 0.003 | 0.35 ± 0.001 | 0.14 |
| Relative weight of kidney | g/100 g bwt | 0.69 ± 0.004 | 0.67 ± 0.002 | 0.028 * |
| **Blood chemistry** | | | | |
| Non-fasting glucose | mmol/l | 7.53 ± 0.07 | 6.94 ± 0.03 | 0.0057 * |
| Insulin | nmol/l | 0.38 ± 0.02 | 0.22 ± 0.005 | 9.6E-5 * |
| Serum TG | mmol/l | 0.36 ± 0.01 | 0.43 ± 0.01 | 0.048 * |
| Total cholesterol | mmol/l | 1.13 ± 0.03 | 1.07 ± 0.01 | 0.39 |
| HDL-C | mmol/l | 1.04 ± 0.03 | 0.97 ± 0.01 | 0.27 |
| NEFA | mmol/l | 0.50 ± 0.01 | 0.61 ± 0.01 | 3.8E-4 * |
| Adiponectin | μg/ml | 2.63 ± 0.36 | 2.33 ± 0.35 | 0.589 |
| Leptin | ng/ml | 3.20 ± 0.23 | 3.31 ± 0.12 | 0.692 |
| **Tissue composition** | | | | |
| TG in liver | μmol/g | 6.49 ± 0.24 | 4.7 ± 0.12 | 0.016 * |
| Cholesterol in liver | μmol/g | 10.93 ± 0.16 | 9.27 ± 0.07 | 0.0014 * |
| TG in heart | μmol/g | 2.29 ± 0.08 | 0.71 ± 0.03 | 2.8E-6 * |
| TG in kidney | μmol/g | 5.33 ± 0.21 | 5.57 ± 0.12 | 0.69 |
| TG in muscle | μmol/g | 1.35 ± 0.09 | 1.29 ± 0.06 | 0.80 |
| **Glucose and lipid metabolism** | | | | |
| Glucose oxidation in BAT | nmol glucose/g/2h | 441.78 ± 19.17 | 386.76 ± 14.01 | 0.39 |
| Basal lipogenesis in BAT | nmol glucose/g/2h | 294.37 ± 11.14 | 262.82 ± 6.63 | 0.34 |
| Basal lipogenesis in epid. fat | nmol glucose/g/2h | 481.05 ± 18.1 | 417 ± 8.62 | 0.80 |
| Insulin-stimulated lipogenesis in epid. fat | nmol glucose/g/2h | 717.14 ± 27.08 | 581.48 ± 23.12 | 0.23 |
| Basal lipolysis in epid. fat | μmol NEFA/g | 3.01 ± 0.05 | 3.51 ± 0.05 | 0.58 |
| Adrenaline-stimulated lipolysis in epid. fat | μmol NEFA/g | 6.09 ± 0.16 | 5.25 ± 0.13 | 0.15 |
| Basal glycogenesis in diaphragm | nmol glucose/g/2h | 892.26 ± 43.12 | 688.17 ± 30.06 | 0.37 |
| Insulin-stimulated glycogenesis in diaphragm | nmol glucose/g/2h | 1554.32 ± 35.04 | 701.84 ± 41.14 | < 1E-4 * |
| Glucose oxidation in diaphragm | nmol glucose/g/2h | 488.38 ± 20.66 | 343.98 ± 8.29 | 0.0042 * |
| **Oxidative stress in the liver** | | | | |
| SOD activity | U/mg | 0.16 ± 0.004 | 0.12 ± 0.003 | 0.026 * |
| GPx activity | μmol GSH/min/mg | 293.14 ± 6.48 | 234.57 ± 5.26 | 0.021 * |
| GR activity | μmol NADPH/min/mg | 132.14 ± 4.51 | 112.57 ± 2.52 | 0.18 |
| CAT activity | μmol $H_2O_2$/min/mg | 1260 ± 45.18 | 1522.14 ± 50.8 | 0.17 |
| GSH/GSSG | | 39.64 ± 1.25 | 25.56 ± 0.97 | 0.0051* |
| GSH | μmol/mg | 75.19 ± 1.18 | 72.94 ± 0.61 | 0.49 |
| GSSG | μmol/mg | 1.96 ± 0.07 | 3.15 ± 0.15 | 0.039 * |
| Conjugated dienes | nM/mg | 35.29 ± 0.88 | 42.86 ± 0.71 | 0.027 * |
| TBARS | nM/mg | 1.73 ± 0.06 | 1.36 ± 0.05 | 0.095 |

Table). We next carried out functional annotations using Ingenuity Pathway Analysis (IPA), which infers shifts in activity of canonical pathways and potential upstream regulators from the

**Table 3. Statistical analysis of response to insulin and adrenaline stimulation in tissues isolated from SHR-*Tti2*$^{+/-}$ rats and SHR wild type littermates.**

| Phenotype | Term/Contrast | Statistic (d.f.) | *p*.value |
|---|---|---|---|
| Glycogenesis in diaphragm | Genotype | $X^2$ (1) = 28.25 | 1.1E-7 * |
| | Insulin | $X^2$ (1) = 9.72 | 0.0018 * |
| | Genotype:Insulin | $X^2$ (1) = 11.03 | 9E-4 * |
| | SHR:250 –SHR:0 | $t$(12) = 4.55 | <1E-4 * |
| | SHR-*Tti2*$^{+/-}$:250 –SHR-*Tti2*$^{+/-}$:0 | $t$(11) = 0.1 | 1 |
| Glucose oxidation in diaphragm | Genotype | $X^2$ (1) = 7.82 | 0.0052 * |
| | Insulin | $X^2$ (1) = 1.34 | 0.25 |
| | Genotype:Insulin | $X^2$ (1) = 3.07 | 0.08 |
| | SHR:250 –SHR:0 | $t$(12) = 2.08 | 0.13 |
| | SHR-*Tti2*$^{+/-}$:250 –SHR-*Tti2*$^{+/-}$:0 | $t$(12) = -0.27 | 0.99 |
| Lipogenesis in epididymal fat | Genotype | $X^2$ (1) = 2.21 | 0.14 |
| | Insulin | $X^2$ (1) = 34.59 | 4.1E-9 * |
| | Genotype:Insulin | $X^2$ (1) = 1.14 | 0.29 |
| | SHR:250 –SHR:0 | $t$(12) = 4.66 | <1E-4 * |
| | SHR-*Tti2*$^{+/-}$:250 –SHR-*Tti2*$^{+/-}$:0 | $t$(12) = 3.75 | 7.7E-4 * |
| Lipolysis in epididymal fat | Genotype | $X^2$ (1) = 0.35 | 0.56 |
| | Adrenaline | $X^2$ (1) = 62.01 | 3.4E-15 * |
| | Genotype:Adrenaline | $X^2$ (1) = 5.08 | 0.024 * |
| | SHR:250 –SHR:0 | $t$(12) = 6.86 | <1E-4 * |
| | SHR-*Tti2*$^{+/-}$:250 –SHR-*Tti2*$^{+/-}$:0 | $t$(12) = 4.48 | < 1E-4 * |

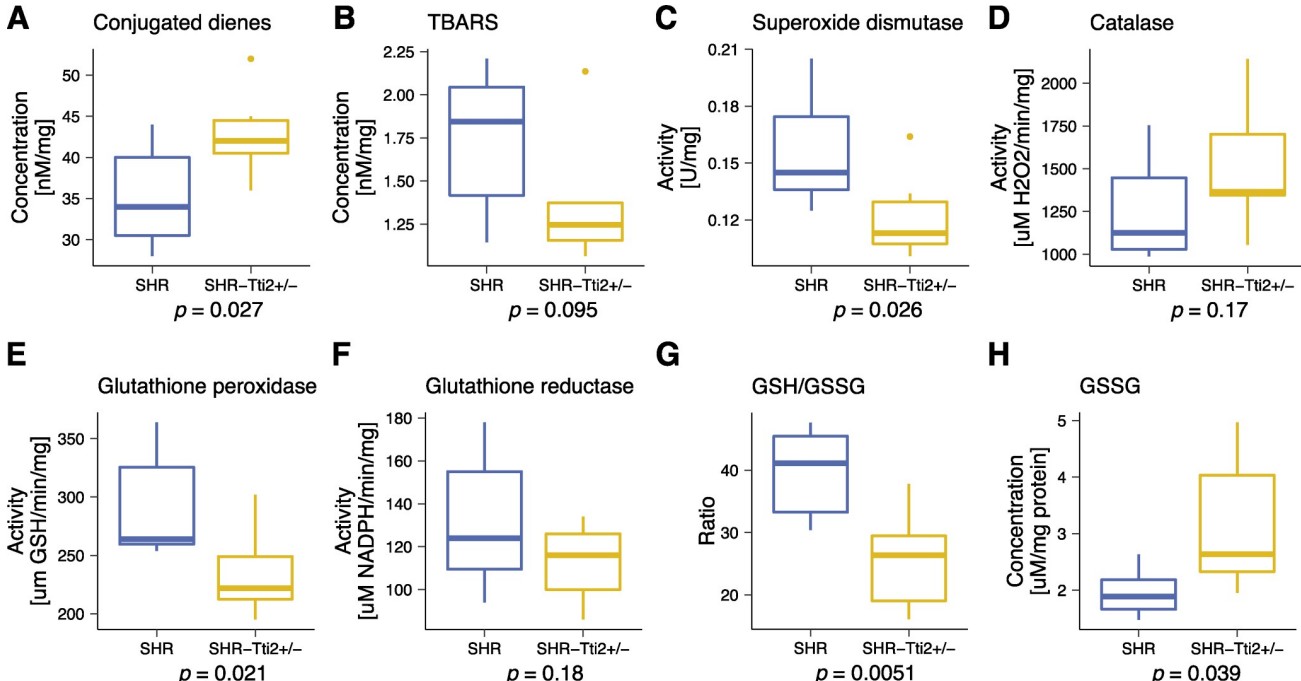

**Fig 4. Knock-down of *Tti2* alters oxidative status in the liver.** Liver extracts prepared from three-month-old heterozygous SHR-*Tti2*$^{+/-}$ rats (N = 7) and wild type SHR littermates (N = 7) were used to measure markers of oxidative stress as indicated in the figure. *p* values were derived from Student's *t*-test. Raw data can be found in S1 Data. Abbreviations: GSH, glutathione; GSSG, oxidised glutathione; TBARS, thiobarbituric acid reactive substances.

direction and magnitude of gene expression changes using curated knowledge databases. Furthermore, IPA can link differentially expressed genes to downstream outcomes.

We first checked whether observed gene expression changes could be connected to altered activity of any of the PIKK. Among canonical pathways, IPA indicated enrichment for genes associated with ATM signalling in the liver, muscle, and fat, and with mTOR signalling in the muscle (S2B–S2D Fig and S2 Data). Additionally, IPA predicted SMG1 as a potential upstream regulator in the liver and hippocampus (S3 Data). The majority of changes in gene expression were not directly connected to PIKK activity.

In agreement with the metabolic phenotype of SHR-Tti2$^{+/-}$ rats, the transcriptional profile of the liver pointed to dysglycemia, specifically to decreased glucose tolerance and increased insulin resistance (Fig 5A and 5B). IPA suggested four upstream regulators that could be linked to glucose intolerance: interleukin 6 (IL6), tuberous sclerosis complex 2 (Tsc2), peroxisome proliferator-activated receptor gamma coactivator 1-beta (Ppargc1b), and lysine (K)-specific histone demethylase 1A (Kdm1a) (S3 Fig). In addition, IPA indicated inhibition of insulin signalling as the most significant upstream regulator (S3 Data). The functional analysis also predicted a broad range of other metabolic changes, for example hepatic steatosis, decreased fatty acid metabolism and lipid synthesis (Fig 5A and 5C and S4 Data). Even though only few differentially expressed genes were detected in the hippocampus of SHR-Tti2$^{+/-}$ rats, these genes indicated decreased cellular homoeostasis (Fig 5A and 5D) and increased apoptosis. Very strong upregulation (24-fold) of sphingosine1-phosphate (S1P) receptor 3 (S1pr3) and downregulation of alkaline ceramidase 2 (Acer2) suggested de-regulation of S1P signalling pathway in the hippocampus (S2A Fig).

## Human genomic variation links *TTI2* expression with genome-wide phenotype associations

Our results indicated that changes in expression levels of *Tti2* can affect both neural and metabolic homeostases. The majority of disease-associated variants in humans are likely to be involved in regulation of transcription [58,59]. Therefore, to link our findings to genomic variation in the human population, we searched the human eQTL catalogue (https://www.ebi.ac.uk/eqtl/) for variants associated with changes in *TTI2* mRNA expression. We extracted 821 variants underlying 2177 eQTL ($p < $ 1e-4) in 28 different tissues and cell types (S5 Data). Out of these, 214 variants were associated with the *TTI2* eQTL in at least 3 distinct tissue types (S4 Fig). The multi-tissue eQTL are more likely to represent true associations as well as being consequential for a wide range of phenotypes [17,18]. We next used all *TTI2* eQTL variants to query the GWAS databases (https://www.ebi.ac.uk/gwas/) for known phenotypic associations. We found 7 variants with reported associations to protein and stem cell factor blood concentrations, blood pressure and frontotemporal dementia (Table 4).

821 variants linked to *TTI2* eQTLs ($p < $ 1E-4) were used to search human genome-wide association studies (GWAS) catalogue. *Data sets* column lists tissues and cell types in which significant eQTLs were detected for a given variant. For GWAS studies for which risk frequencies were not reported, we included variant frequencies from phase 3 1000 Genomes Project combined population (denoted with asterisks).

## Discussion

In this study we used systems genetics methods to i) discover a number of metabolic and endocrine genetic correlates of the two critical parameters of adult hippocampal neurogenesis—precursor cell proliferation and net neurogenesis; and ii) identify *Tti2* as a molecular link between brain cellular plasticity and peripheral metabolism.

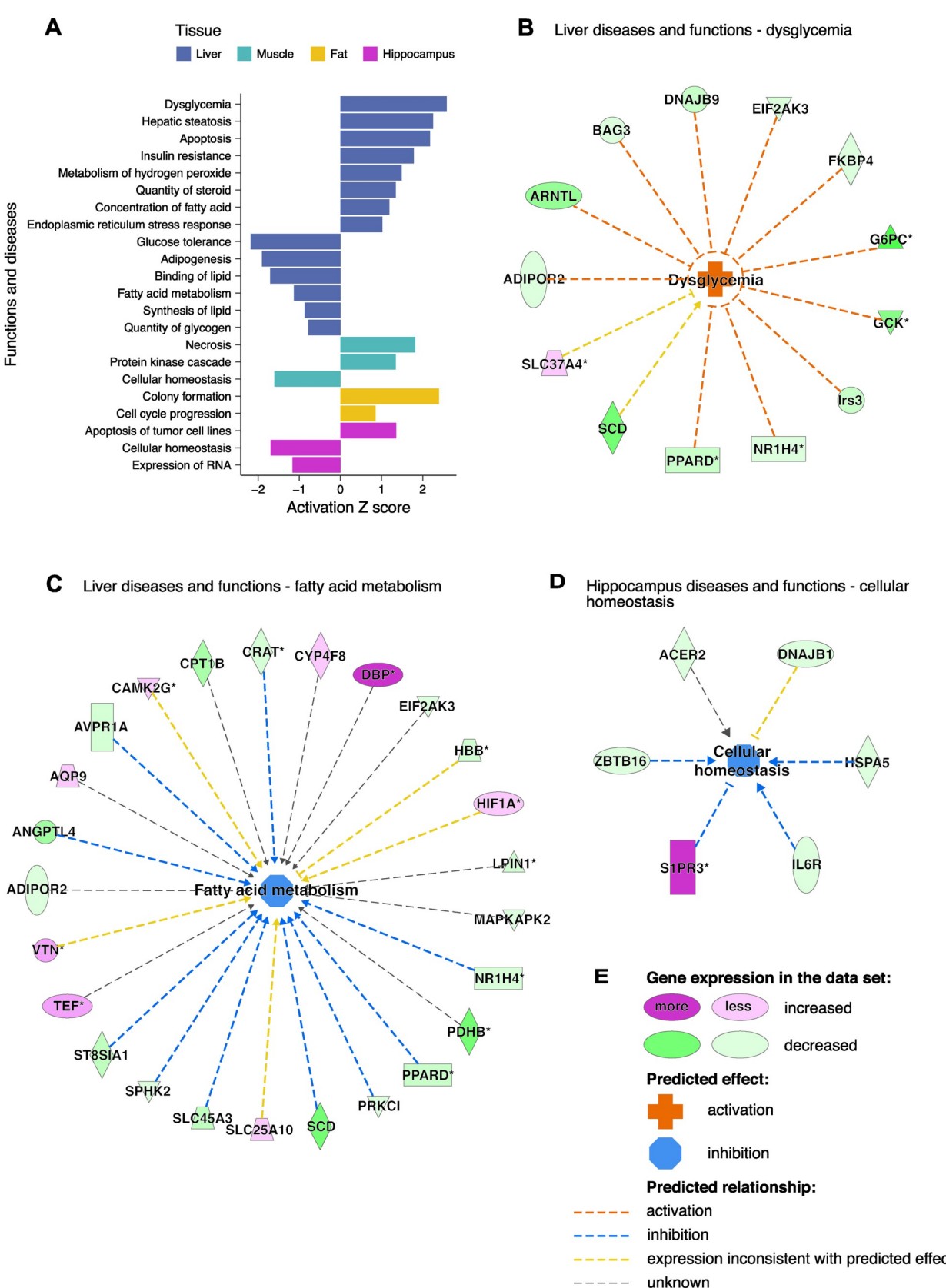

**Fig 5. Functional analysis of gene expression changes indicates insulin resistance and loss of cellular homeostasis in SHR-*Tti2*^+/- rats.**
Differentially expressed genes between SHR-*Tti2*^+/- rats and wild type SHR littermates in the liver, muscle, perirenal adipose tissue and hippocampus were analysed using Ingenuity Pathway Analysis (IPA). IPA predicts activation or inhibition of functions, pathological processes and molecular pathways from the direction and magnitude of expression changes using curated databases. (A) Graph depicts activation (positive Z-score) or inhibition (negative Z-score) of top functions and diseases in each of the analysed tissues. Full list of significantly affected functions can be found in S4 Data. (B-D) Edges in each network illustrate predicted relationships between upregulated (magenta) and downregulated (green) genes and downstream functions (centre nodes; orange, activating effect; blue, inhibitory effect). (E) Colour key. Asterisks in network graphs denote multi-protein complexes.

## The HXB/BXH family is well-suited to discover genetic physiological correlates of cognitive endophenotypes

The HXB/BXH family of 30 members has only modest statistical power to detect QTL, as the majority of loci have only small effects on phenotypes and variants behind a substantial part of heritability fall below stringent significance levels (power calculations to detect a QTL for a population of 30 strains for different heritability and effect size values are presented in S5 Fig). Nonetheless, the chief advantage of recombinant inbred strains is their suitability to measure the tendency of traits to co-segregate [47]. Because each inbred line can supply an indefinite number of isogenic individuals, multiple phenotypes can be measured in the same genotype, the collected data are cumulative and comparable across time and laboratories. Use of separate cohorts of animals to assess different phenotypes avoids intra-individual correlations (which may, for example, be due to the health status of a particular animal) and ensures that any associations between traits are due to shared genetic variation. Genetic correlations, therefore, may suggest the presence of allelic variants with pleiotropic effects on correlating phenotypes [12,20]. Here we show that, under baseline laboratory conditions, precursor cell proliferation and the final outcome of the process of adult neurogenesis—the numbers of surviving and functionally integrated neurons—are under the control of largely distinct sets of genes. Accordingly, each of these traits correlated to non-overlapping collections of metabolic and endocrine parameters. Because adult neurogenesis—although a complex process in itself—represents one specific aspect of hippocampal plasticity, these correlations advance our understanding of interactions between global metabolic features and brain function. Here we proceeded to dissect in-depth the correlation between serum glucose and net neurogenesis, which could both be mapped to a common overlapping QTL. The majority of discovered correlations, however, could not be explained by such a strong genetic association, in agreement

**Table 4. Human genomic variants with overlapping associations to *TTI2* expression and phenotypes.**

| Variant ID | eQTL beta direction | eQTL data sets | Associated trait | Trait ID | GWAS $p$ value | GWAS beta direction | Risk Frequency* |
|---|---|---|---|---|---|---|---|
| rs10094645 | increase | Artery_Aorta, Muscle_Skeletal, Testis, Thyroid, Whole_Blood | blood protein measurement | EFO_0007937 | 4E-23 | decrease | 0.594 |
| rs2732317 | increase | Artery_Aorta, Testis, Thyroid, Whole_Blood | blood protein measurement | EFO_0007937 | 3E-77 | decrease | 0.612 |
| rs2732260 | increase | Monocyte | frontotemporal dementia, memory impairment | Orphanet_282 EFO_0001072 | 1E-6 | NA | 0.03 |
| rs2732259 | increase | Esophagus_Muscularis, Muscle_Skeletal, Testis, Whole_Blood | hypoxanthine measurement | EFO_0010500 | 7E-6 | increase | 0.619* |
| rs6996562 | increase | Artery_Aorta, Testis, Whole_Blood | pulse pressure measurement | EFO_0005763 | 2E-10 | increase | 0.473 |
| rs7845722 | increase | Artery_Aorta, Muscle_Skeletal, Testis, Whole_Blood | pulse pressure measurement | EFO_0005763 | 1E-9 | increase | 0.4 |
| rs1568119 | decrease | Monocyte | stem cell factor measurement | EFO_0008291 | 1E-7 | increase | 0.08* |

with the polygenic nature of quantitative traits. Other methods, such as gene co-expression analyses, have the potential to integrate available data and elucidate mechanisms underlying remaining relationships [35].

## *Tti2* as a pleiotropic gene regulating net neurogenesis and metabolic homeostasis

Using transcriptional profiles from peripheral tissues and the hippocampus as intermediate phenotypes between complex physiological outcomes and genomic variation we were able to narrow down the 'serum glucose-neurogenesis' QTL interval to the *Tti2* candidate gene. Tti2, together with its binding partners, Tti1 and telomere maintenance 2 (Telo2) protein, form the Triple T (TTT) complex [60], which associates with a number of molecular chaperones, including Hsp90, Hsp70, Hsp40, and the R2TP/prefoldin-like complex [61,62]. Interaction of TTT with R2TP modifies ATPase activity of R2TP components, RUVBL1-RUVBL2 [63,64]. TTT binds to nascent peptides of PI3K-related protein kinases (PIKK) and delivers them to the R2TP chaperone, thereby acting as a scaffold adaptor and a critical regulator of PIKK abundance in mammalian and yeast cells [60,62–67]. In mammals, the PIKK family consists of ATM, ATR, mTOR, TRAPP, SMG1, and DNA-PKcs. These proteins play strategic roles in multiple cellular functions, such as genome stability, DNA repair, regulation of gene and protein expression, nonsense-mediated RNA decay, cell growth and cell cycle progression, and regulation of responses to nutrient availability [68–71]. All PIKK are also an important part of stress responses [72].

In our present study, we generated a heterozygous SHR-*Tti2*$^{+/-}$ line carrying a frame-shift mutation, which resulted in a series of premature stop codons in the N-terminal domain of *Tti2*. Cryo-electron microscopy structure of the human R2TP-TTT complex has revealed that α-helical regions in the N-terminal domain of Tti2 form an interface for interaction with Tti1 and RUVBL, and are essential for the formation and function of TTT [63,64]. Accordingly, mutational screens in yeast produced viable cells only when truncations were located at the very end of the C-terminus of *Tti2* [73]. Therefore, we predicted that this modification results in a non-functional protein. *Tti2* mRNA levels in the heterozygous SHR-*Tti2*$^{+/-}$ rats were reduced by half compared to wild type littermates, as established by RNA sequencing, implicating nonsense-mediated decay of the mutated mRNA. We used heterozygous animals because our aim was to mirror the eQTL effect and reduce the amount of available Tti2 rather than remove it completely. The SHR background had been chosen because the SHR allele is associated with higher *Tti2* expression. Although antibodies recognising rodent Tti2 were not available, experiments in yeast have suggested that Tti2 protein abundance is directly correlated to mRNA expression [60]. SHR-*Tti2*$^{+/-}$ rats had similar body weights, gross morphology and general cage behaviour as the wild-type littermates. However, reducing *Tti2* expression by half led to a reduction of the number of new neurons in the DG, a concomitant lowering of serum glucose and insulin concentrations, and to hallmarks of insulin resistance in skeletal muscles. Metabolic alterations in SHR-*Tti2*$^{+/-}$ rats went beyond glucose homeostasis: we also found changes in lipid metabolism and elevated oxidative stress in the liver. Together, these findings support *Tti2* as a causal gene within the joint 'serum glucose-neurogenesis' QTL.

Expression of *Tti2* correlated negatively to adult neurogenesis and serum glucose concentrations in the HXB/BXH family, yet in the SHR rats with only one functional copy of the *Tti2* gene we saw a further decrease of each trait value. This disparity would imply that the SHR allele is associated with decrease of the Tti2 protein function despite higher *Tti2* mRNA abundance. The SHR allele carries six amino-acid substitutions, including Glu247Asp within the highly conserved Tti2 family super-helical central domain (conservation score 1.0; S1 Table)

and Lys198Glu at a moderately conserved residue (score 0.57) in the N-terminal portion of the protein. Although all substitutions were scored as benign by prediction algorithms SIFT and PolyPhen, it cannot be excluded that they have an impact on protein stability or interactions with any of the binding partners. Most of Tti2 consists of α-helical HEAT repeats and large regions of the protein are involved in interactions with Tti1, RUVBL and PIKKs [63,64]. Higher mRNA expression could evolve independently or as a compensation of reduced function. For example, in the duplicated maize genome, the number of copies of genes encoding TTT complex members and PIKK have all reverted to one, suggesting evolutionary pressure to maintain gene dosage balance [74].

Transcriptome profiling of SHR-*Tti2*[+/-] rats revealed extensive changes in gene expression in the liver, and to a lesser degree in skeletal muscle, perirenal fat and hippocampus compared to SHR wild type littermates. Liver, together with skeletal muscle and adipose tissue, are decisive organs in maintaining glucose homeostasis and, hence, the development of insulin resistance [75]. Functional analysis of differentially expressed genes in the liver identified networks of genes and potential regulators whose activation and inhibition could explain insulin resistance and dysglycemia in the heterozygous animals. We also recorded significant upregulation of *Insr* in the muscle, which IPA interpreted as consistent with hypoglycaemia and insulin resistant diabetes (S4 Data).

We also used IPA to predict which upstream regulators could be activated or inhibited in a manner consistent with observed gene expression changes. The vast majority of differentially expressed genes were not linked to PIKK activity. Thus far only PIKK peptides were identified as clients of the TTT complex despite attempts to capture other target proteins [73]. Therefore it is unlikely that another, yet unknown, pathway contributes to the observed phenotypes. Depletion of Tti2 destabilises all PIKK proteins and impairs nuclear localisation of ATM, ATR and TRRAP, but does not affect their mRNA abundance [60,73]. Notwithstanding the lack of direct evidence, reduction of *Tti2* expression in SHR-*Tti2*[+/-] rats may destabilise the TTT complex and consequently impair signalling of PIKK. It has been reported that reduced PIKK signalling due to tissue-specific targeting of selected genes in the mouse led to impaired adipogenesis (reduced fat deposits), insulin resistance with lower insulin-stimulated glucose transport, reduced antilipolytic effects of insulin (increased NEFA levels), and ectopic fat accumulation [76–80]. These results are similar to metabolic disturbances observed in SHR-*Tti2*[+/-] rats suggesting involvement of the same molecular pathways. Our data also do not allow differentiating whether all or only some of the PIKK are compromised in the SHR-*Tti2*[+/-] rats. Destabilisation of the TTT complex has strongest effects on ATM and ATR protein levels and to a lesser extent on other PIKK [60,67,73,81]. Interestingly, in SHR-*Tti2*[+/-] rats IPA detected enrichment of differentially expressed genes related to ATM signalling in the liver, muscle and fat and mTOR signalling in the muscle. While these results do not imply that other PIKK were not affected, ATM might be the most sensitive to Tti2 downregulation also in rats used in our study.

## Direct vs. metabolic *Tti2*-dependent regulation of neurogenesis

Pleiotropy occurs when a single genomic variation, or more broadly a change in a function of a single gene, has multiple consequences at the phenotypic level [9]. Because metabolic diseases can, to some extent, be modified by lifestyle interventions in order to prevent or dampen cognitive decline, but genes cannot, it is clinically crucial to understand which correlations between metabolic and cognitive phenotypes arise from genetic predisposition due to pleiotropic genes. Because we used targeted mutagenesis at the *Tti2* locus rather than tissue specific approaches to confirm association with target phenotypes, we cannot exclude that Tti2 affects

neurogenesis through circulating metabolites or hormones. Correlations between traits in the absence of genetic variation indicate indirect effects [10].

Hippocampal plasticity and neurogenesis are intricately related to nutrient availability and insulin and insulin-like growth factor 1 (IGF1) signalling [82–85]. Glucose is the primary energy source for the nervous system. Hyperglycemia and insulin resistance are detrimental to the brain and negatively influence adult hippocampal neurogenesis [86–89]. In addition, caloric restriction, which lowers plasma insulin and glucose levels, resulted in increased survival of new neurons and higher net neurogenesis [90]. These associations have the opposite effect to the genetic correlation recorded in the HXB/BXH family and hint that, as such, lower serum glucose measured in SHR-*Tti2*$^{+/-}$ rats may not necessarily lead to lower survival of adult-born neurons. In the HXB/BXH family, neurogenesis correlated positively to serum insulin (Pearson's $r = 0.35$, $p = 0.09$). However, we did not detect any consistent associations with measures of insulin resistance, suggesting that peripheral insulin resistance is also unlikely a sole cause of neurogenesis impairment in SHR-*Tti2*$^{+/-}$ rats. Furthermore, transcriptional profiling of the hippocampus from heterozygous animals did not indicate brain insulin resistance.

Metabolic tissues in heterozygous rats also manifested deregulated lipid metabolism. Higher levels of circulating triglycerides and free fatty acids could further contribute to disrupted glucose metabolism and neurogenesis. For example, high fat diets, which increase circulating plasma lipids and lipid peroxidation, have been documented as detrimental to neurogenesis and cognition [91,92]. On the other hand, serum concentrations of two major adipose tissue hormones—leptin and adiponectin—were similar in SHR-*Tti2*$^{+/-}$ and intact animals.

The support in favour of direct effects of Tti2 reduction on neurogenesis comes from the severe neural deficits in individuals carrying loss-of-function mutations in *TTI2* in the absence of serious metabolic insufficiencies. In humans, homozygous and compound heterozygous loss-of-function mutations in *TTI2* cause microcephaly, severe intellectual disability, dysmorphic facial features, short stature, speech and movement disorders, and skeletal deformations [81,93–95]. Similar abnormalities were observed in children carrying *TELO2* mutations [96,97]. Particularly, failures of DNA repair pathways downstream of the TTT complex have detrimental effects on development and maintenance of the central nervous system. Recessive mutations in ATM cause ataxia telangiectasia, a disease characterised by progressive neuronal degeneration [98,99]; while loss of ATR leads to Seckel syndrome characterised by postnatal dwarfism, microcephaly, intrauterine growth defects, and mental retardation [100]. ATM, ATR and DNA-PKcs are essential to preserving the genome integrity during replication [101] and thus their function is particularly important in dividing precursor cells in the course of neurogenesis [102], also in the adult hippocampus [103]. Knock-out of ATM or ATR in mice has detrimental effects on brain development, with pronounced loss of hippocampal neurons [104]. ATM-deficient mice have abnormally increased rates of proliferation with concomitantly lowered survival of new neurons [103]. In human neural stem cells, ATM suppresses excessive retrotransposition [105], the process which contributes to neural diversity and plasticity during hippocampus development, and then in the adult stem cells [106].

mTOR also plays multiple roles in the development and function of the brain [107], including maintenance of neural progenitor cell pools. During embryonic development, conditional knockout of mTOR in neural stem cells dramatically reduced their proliferation thereby reducing production of postmitotic neurons and brain size [108]. Similarly, inhibition of mTORC1 signalling in the neural stem cells in the neonatal subventricular zone (SVZ) of the lateral ventricle, which also harbours a population of neural stem cells that persist throughout life, reduced generation of transient amplifying precursor cells and thus decreased the abundance of their differentiated progeny [109]. mTOR signalling promoted survival of neural stem cells isolated from SVZ *in vitro* downstream of EGFR signalling [110]. In addition,

transient systemic inhibition of the mTOR pathway by rapamycin in early postnatal life resulted in abnormal proliferation, reduced progenitor cell numbers, and eventually decreased the volume of the adult dentate gyrus [111]. Similarly, treatment with rapamycin in adult mice also reduced proliferation of hippocampal progenitor cells and net neurogenesis [112]. Rapamycin also prevented increase of neurogenesis by physical activity, which is a known pro-neurogenic stimulus [112].

ATM and mTOR are both downstream targets of insulin and IGF1 signalling [98]. Insulin and IGF1 provide trophic signals that can both stimulate and inhibit proliferation and survival of adult precursor cells *in vivo* [83,113–118]. Insulin is also expressed directly in neuronal progenitors [119]. The massive upregulation of S1pr3 and downregulation of Acer2 in the hippocampus of SHR-*Tti2*$^{+/-}$ rats suggested changes in the sphingosine-1-phosphate (S1P) signalling pathway, which is implicated in the control of cell death and survival, as well as synaptic plasticity via interactions with multiple cellular signalling cascades [120–123]. Interestingly, S1pr3 potentiates IGF1 signalling [124], and cross-links with the mTOR-AKT nutrient sensing pathway [125]. Although our analysis of differentially expressed genes did not suggest changes in PIKK activity in the hippocampus of heterozygous animals, the precursor cells and immature neurons are only a minor fraction of the entire hippocampus and therefore we might not have captured genes affected specifically in these cells. For example, Ka and colleagues [108] found that in the developing cerebral cortex mTOR signalling was detected mostly in the radial neural stem cells, the principal precursor cells of the developing central nervous system. mTOR activity in the postmitotic neuronal layers was very low. In the early postnatal and adult SVZ, mTOR activity was also concentrated in actively proliferating transient amplifying progenitor cells [109,126]. Compared to dividing precursor cells, expression of ATM and ATR is also downregulated in post-mitotic neurons, where these proteins are involved in clathrin-mediated endocytosis at synapses [127].

All told, the interwoven relationships between peripheral metabolism, insulin and PIKK signalling pathways point to complex responses to intracellular deficits in PIKK and extracellular signals in the brain of SHR-*Tti2*$^{+/-}$ rats, and lend support for truly pleiotropic roles of *Tti2* in the regulation of glucose homeostasis and structural brain plasticity.

## Limitations

In our study we used only male rats and thus we do not know whether any of the identified associations interacts with sex. Adult neurogenesis has a strong environmental component, where variability between individuals depends on the experience and life history. To avoid interference of systemic environmental errors with genetic sources of variation and obtain reliable estimates of neurogenesis in the HXB/BXH family, we sampled several individuals per strain from at least 3 independent litters born at different time points throughout the data collection period. Inclusion of both sexes may have increased within-strain variation, therefore decreasing power to detect genetic associations. To use sex as a covariate would require a larger number of experimental animals, which was not logistically feasible. Under these circumstances, we chose to use males for compatibility with existing data, as the vast majority of published HXB/BXH phenotypes were measured in young male rats to avoid inter-individual variation due to oestrous cycle. The inbred status of the HXB/BXH family enables integration of additional data at later time points.

Another limitation of our study was the use of a general rather than tissue-specific *Tti2* knockout. This choice was dictated by the complexity of both processes under study. Glucose homeostasis is achieved by interactions between many cell types in several tissues. Adult neurogenesis is not only regulated intrinsically in neural stem cells, but is also influenced by the

niche in which they are harboured, and network activity of the entire hippocampus. A classical knockout approach also reflected the presence of *Tti2* cis-eQTL in all examined tissues. Although we cannot unequivocally claim that the effect of *Tti2* knockout on neurogenesis is independent of metabolism, we have discussed literature that support the hypothesis of a direct, horizontal pleiotropic effect of *Tti2* dosage on these distinct phenotypes.

## Conclusion

Understanding of the molecular events by which a genomic variation leads to physiological consequences for the organism across many functions provides a foundation for effective precision medicine. Our study exemplifies the power of rodent genetic reference populations not only to identify associations between phenotypes that are difficult or even impossible to assess in humans, but also to give insights into the cell biology behind these associations. Our experiments showed that manipulating the abundance of a single component of the protein folding machinery had relatively subtle yet significant effects on a broad range of phenotypes. Mining human data sets revealed more than 800 genomic variants that are linked to *TTI2* expression, seven of which refer to associations with protein and blood stem cell factor concentrations, blood pressure, and frontotemporal dementia. Given the dose-dependent effects of *Tti2* on adult hippocampal neurogenesis and glucose homeostasis, we speculate that human variants that affect *TTI2* expression or function may also have quantitative effects on these phenotypes.

## Materials and methods

### Ethics statement

All experiments were performed in agreement with the Animal Protection Law of the Czech Republic and were approved by the Ethics Committee of the Institute of Physiology, Czech Academy of Sciences, Prague (Permit number: 66/2014).

### Animals

Brown Norway BN-*Lx*/Cub, spontaneously hypertensive SHR/OlaIpcv (referred to as BN and SHR, respectively) and 30 HXB/BXH recombinant inbred strains, as well as SHR-*Tti2*[+/-] knockout heterozygous rats used in the current study were housed in an air-conditioned animal facility at the Institute of Physiology, Czech Academy of Sciences. HXB/BXH strains are inbred for more than 80 generations. Animals were maintained on a 12 h light/dark cycle in standard laboratory cages provided with standard laboratory chow and water *ad libitum*. To assess survival of new-born cells in the dentate gyrus, 10-week-old animals were given 3 daily intraperitoneal injections of 50 mg/kg bromodeoxyruidine (BrdU; Sigma) and perfused 28 days later. We studied 5–9 male rats from each strain derived from at least 3 independent litters (total 243 rats). To isolate tissues for RNA and protein isolation, rats were anaesthetised with ketamine and decapitated. Tissues were placed in RNA later (microarray) or snap frozen in liquid nitrogen. For the microarray analysis, one male and one female 10-week-old rat from each parental and HXB/BXH strains were used (total 64 rats). Biochemical, metabolic and hemodynamic phenotypes were assessed in three-month-old non-fasted male SHR-Tti2[+/-] rats and their wild-type littermates (N = 8 per strain).

### Generation of Tti2 knockout SHR rats

*Tti2* knockout rats were generated by microinjecting fertilized ova of SHR rats with the ZFN (Zinc Finger Nuclease) construct from Sigma-Aldrich. The construct was designed to target the first exon using the following sequence of ZFN binding (capital letters) and cutting site

(small letters): TCTGACCCGGATCCAAGCaccaagGGTGGGTGGCAGGGC. DNA samples isolated from 452 rats born after microinjection with ZFN construct were amplified using primers flanking the target sequence: ZFN F: 5'-TACACTGTGATTGGCTGGGA-3' and ZFN R: 5'-GGCGCAGTGGAGTGATC-3'. Altogether 4 positive animals were detected. An SHR-*Tti2*[tm1]/Ipcv line no.14 (referred to as SHR-*Tti2*[+/-]) with an 8 bp deletion (NM_001013883.1 (Tti2):c.243_250delCGAGATCC; on the protein level: NP_001013905.1:p.Glu82Glyfs) has been selected for further analyses. The heterozygous founder was crossed with SHR and F1 rats were intercrossed. SHR-*Tti2*[+/-] heterozygotes were selected for breeding and phenotyping while their wild type littermates were used as controls.

## Histology

Histology was carried out using standard procedures [48]. Rats were deeply anaesthetised with a mixture of ketamine/xylazine and intracardially perfused with 0.9% NaCl, followed by ice-cold 4% paraformaldehyde (PFA) in 0.1 M phosphate buffer (PB), pH 7.4. The brains were removed, post-fixed overnight in 4% PFA and equilibrated in 30% w/v sucrose in 0.1 M phosphate buffer. 40 μm frozen coronal sections were cut on a sliding microtome (Leica) and stored in a cryo-protective solution (25% ethylene glycol, 25% glycerol in 0.1 M PB) at –20°C.

For immunohistochemistry, every sixth section was washed in Tris-buffered saline (TBS) and pretreated in 3% $H_2O_2$ and 10% methanol in TBS for 15 min. After several washes in TBS and NaCl, DNA was denatured with 2.5 M HCl for 30 min at 37°C. Multiple washes in phosphate-buffered saline (PBS) were performed between each subsequent step. The sections were incubated in a blocking solution containing 10% donkey serum and 0.3% Triton-X100 in PBS for 1 h and then with primary antibodies (rat anti-BrdU, 1:500, OBT0030, AbD Serotec; or rabbit anti-Ki67, 1:500, NCL-Ki67p, Novocastra) diluted in the blocking solution for 48–72 h at 4°C. Sections were incubated for 2.5 h with biotinylated secondary antibodies (1,500, Jackson ImmunoResearch) diluted in the blocking solution containing 3% donkey serum. Immunocomplexes were detected using the Vectastain Elite ABC kit (Vector Laboratories) and 0.02% diaminobenzidine (D5905, Sigma) enhanced with $NiCl_2$. After mounting in 0.1 M PB onto gelatine-coated glass slides, the sections were air dried, cleared in the alcohol gradient series, and coverslipped with Entellan New (Merck).

Cells were quantified using a simplified optical fractionator method as discussed before [48,128]. BrdU positive cells were quantified along the rostro-caudal axis in the granule cell layer (GCL) and subgranular zone (SGZ) defined as a two-cell wide band below the GCL. Ki67 cells were quantified in the three-cell wide zone below the GCL and in the inner third of the GCL. Clusters of cells were defined as at least three cells not further apart than two-cell diameter. The cells in the uppermost focal plane were ignored to avoid oversampling errors. The counts from left and right sides of the DG in each sample were averaged and multiplied by 6 (section sampling interval) to obtain total numbers of newborn (BrdU) or proliferating (Ki67) cells per dentate gyrus.

To assess the identity of BrdU-positive cells and estimate the range of neuronal and astrocyte survival across the HXB/BXH family, sections from three randomly selected individuals from each parental strain and one from each HXB/BXH member were processed for fluorescent staining with BrdU, NeuN and S100β. Every twelfth section was washed twice in NaCl, pretreated in 2 M HCl for 30 min at 37°C and washed in PBS. After blocking, sections were incubated for 48 h at 4°C with primary antibodies (rat anti-BrdU, 1:500, OBT Serotec; mouse anti-NeuN, 1:200, MAB377, Millipore; rabbit anti-S100β, 1:2000, Ab41548, Abcam) diluted in the blocking solution. After washing in PBS, sections were incubated for 4 h with secondary antibodies (anti-rat Cy3, anti-mouse DyLight 488, anti-rabbit Cy5, 1:500, Jackson

Immunoresearch) diluted in the blocking solution, washed in PBS and mounted onto glass slides using fluorescence mounting medium Aqua-Poly/Mount (Polysciences). 100 randomly selected BrdU-positive cells from each animal along the rostro-caudal axis of the DG were imaged at 400× magnification with spectral confocal microscope (TCS SP2, Leica) and examined for NeuN and S100β immunoreactivity.

## Microarray analysis

Hippocampi were dissected from brains stored in RNA later and RNA was isolated using RNA STAT-60 kit (Tel-Test Inc). RNA was purified using standard sodium acetate-ethanol precipitation method. RNA purity and concentration was evaluated using 260/280 nm absorbance ratio and the quality was checked using Agilent Bioanalyzer 2100 prior to hybridisation. Samples were hybridised onto GeneChip Rat Exon 1.0 ST microarrays (Affymetrix) using manufacturers protocols.

Along with the hippocampus gene expression data, we analysed published data sets from parental and 29 recombinant inbred strains from adrenal gland, liver, skeletal muscle, perirenal fat, kidney, aorta, and ventricle [40,41,43]. These data sets consisted of microarray analysis on Affymetrix Rat230_2 (muscle, liver, aorta and ventricle) and RAE230A (adrenal gland, fat and kidney) chips. Unprocessed microarray expression data were retrieved from ArrayExpress, (www.ebi.ac.uk/arrayexpress; adrenal gland, E-TABM-457; liver, E-MTAB-323; muscle, E-TABM-458; fat and kidney, E-AFMX-7; aorta, E-MTAB-322; left ventricle, E-MIMR-222).

Probes from each data set were assembled into probesets mapped to Ensemble gene identifiers from Rnor_5.0 rat genome release using Version 10 custom cell definition files from the Brain Array (University of Michigan) website [129]. Probes that mapped to regions containing insertions, deletions or single nucleotide polymorphisms in the SHR/Ola or BN-*Lx*/Cub strains compared to the reference genome [46] were removed prior the analysis to avoid spurious linkage due to differential hybridisation. Probesets which after filtering contained less than 3 probes were removed. Gene expression summaries were derived using robust multichip average (RMA) algorithm [130] in the R Affy package [131].

## Heritability

In repeated sampling of isogenic individuals, the variance observed within the genotype can be attributed to environmental influences, whereas differences between strains are primarily due to differences in genotypes. Thus, we defined narrow sense heritability as the intraclass correlation coefficient obtained from a mixed linear model employing restricted maximum likelihood approach using function *lmer* from lme4 R package [132,133].

## QTL mapping

Phenotype QTL were calculated using strain means for surviving (BrdU$^+$) and proliferating (Ki67$^+$) cells in the DG as well as for phenotypes correlating to neurogenesis traits. Expression QTL (eQTL) were calculated for all genes in each data set. Marker regression against SNP-based genotype markers mapped to Rnor_5.0 genome assembly with 2049 unique strain distribution patterns in 29 HXB/BXH strains [134] was performed using the QTLReaper program [135], which reports likelihood ratio statistic (LRS) score at each marker. LRS was converted to logarithm of odds ratio (LOD) by dividing by 4.61, where LOD $\approx -log_{10}(p)$. Empirical genome-wide significance of linkage was determined by a permutation test as previously described [40]. Genome wide significance was defined as the 95$^{th}$ percentile of the maximum LOD score and less stringent suggestive threshold as the 37$^{th}$ percentile, which on average yields one false positive per genome scan [51,52]. The 95% confidence intervals for QTL were

calculated in R/qtl using Bayesian method [136]. Traits with overlapping QTL were summarised as their first principal component, an eigenphenotype (adult neurogenesis and serum glucose) or eigengene (*Tti2* expression), using WGCNA R package [137]. eQTL were defined as local when position of a QTL mapped within 10 Mb from the physical location of the gene [40]. Conditional genome scans were carried out in R/qtl as described previously [50,53,54] to establish relationships between genomic loci, gene expression and phenotypes using function *scanone* with parameter *addcovar*. The gene was considered causal when the LOD score of the phenotype QTL fell below suggestive level after conditioning on its expression. QTL power analysis was carried out with R/qtlDesign R package using function *powercalc* [138]. Analysis of mutations of candidate genes was performed using tools in Rat Genome Database [139]. All genomic positions were mapped to Rnor_5.0 genome assembly.

## RNA isolation and quantitative PCR

RNA was isolated using RNeasy Mini Kit (Qiagen) following manufacturer's instructions. Genomic DNA was removed by on-column DNase digestion. Frozen tissues were homogenised in QIAzol with TissueRuptor (Qiagen). cDNA was synthesised with SuperScript II reverse transcriptase (Invitrogen) using oligo(dT) primers and 1 μg of total RNA. Quantitative PCR was performed using SYBR Green PCR kit (Qiagen) on cDNA corresponding to 25 ng of total RNA using the following conditions: 95˚C for 15 min, and 40 cycles at 94˚C for 15 s, 60˚C for 30 s and 72˚C for 30 s. Gene-specific primer pairs were designed using Primer3 software [140]. Dissociation analysis from 55˚C to 90˚C of the end product was performed to ensure specificity. Cycle of threshold (CT) values were normalised to the GAPDH reference to calculate the relative level of gene expression on the $\log_2$ scale (ΔCT). Mean ΔCT values from BN rats were then subtracted from each sample ΔCT to obtain ΔΔCT values.

## RNA sequencing and analysis

RNA was isolated as described above from the frozen hippocampus, liver, soleus muscle and perirenal adipose tissue samples isolated from SHR-Tti2$^{+/-}$ rats and wild-type SHR-Tti2$^{+/+}$ littermates. RNA integrity was confirmed using BioAnalyzer (Agilent Technologies, Germany). Sequencing libraries were prepared using NEBNext Ultra II Directional RNA Library Prep Kit for Illumina from 300 ng of total RNA, with mRNA enrichment by poly-dT pull down using the NEBNext Poly(A) mRNA Magnetic Isolation Module (NEB) according to the manufacturer's instructions. Samples were then directly subjected to the workflow for strand-specific RNA-Seq library preparation (Ultra II Directional RNA Library Prep, NEB). For ligation custom adaptors were used (Adaptor-Oligo 1: 5'-ACA CTC TTT CCC TAC ACG ACG CTC TTC CGA TCT-3', Adaptor-Oligo 2: 5'-P-GAT CGG AAG AGC ACA CGT CTG AAC TCC AGT CAC-3'). After ligation, adapters were depleted by an XP bead purification (Beckman Coulter) adding the beads solution in a ratio of 1:0.9. Dual indexing was done during the following PCR enrichment (12 cycles, 65˚C) using custom amplification primers carrying the same sequence for i7 and i5 index (Primer 1: AAT GAT ACG GCG ACC ACC GAG ATC TAC AC NNNNNNNN ACA TCT TTC CCT ACA CGA CGC TCT TCC GAT CT, Primer 2: CAA GCA GAA GAC GGC ATA CGA GAT NNNNNNNN GTG ACT GGA GTT CAG ACG TGT GCT CTT CCG ATC T). After two more XP bead purifications (1,0.9), libraries were quantified using the Fragment Analyzer (Agilent). For Illumina flowcell production, samples were equimolarly pooled and sequenced 75bp single-end on multiple Illumina NextSeq 500 flowcells, aiming for approximately 30 million sequencing reads per sample.

Expression levels of individual transcripts were estimated by kallisto (ver. 0.46.1) [141] using Ensembl cDNA database (release 97) [142] as a reference. The software was executed

with sequence-based bias correction, 100 bootstrap samples, an average fragment length of 200 bp and standard deviation set to 20. Differential expression analysis was carried out using an R package sleuth (ver. 0.30.0) [143]. A single outlier hippocampus sample was identified by examining a principal component projection and removed from further analysis. Samples obtained from each tissue were used to fit independent statistical models, with genotype as a single covariate, using 5 or 4 replicates per genotype. Statistical significances of changes in transcript abundances were computed using a Wald test. A 10% FDR (false discovery rate) cut-off and an absolute value of fold change of 1.5 were used to identify differentially expressed genes. Functional enrichment analysis was performed with Ingenuity Pathway Analysis software (IPA; Qiagen) at default settings [144].

## Basal and insulin stimulated glycogen synthesis in skeletal muscle

For measurement of insulin stimulated incorporation of glucose into glycogen, diaphragmatic muscles were incubated for 2 h in 95% $O_2$ + 5% $CO_2$ in Krebs-Ringer bicarbonate buffer, pH 7.4, containing 0.1 µCi/ml of $^{14}$C-U glucose, 5 mmol/L of unlabelled glucose, and 2.5 mg/ml of bovine serum albumin (Armour, Fraction V), with or without 250 µU/ml insulin. Glycogen was extracted, and basal and insulin stimulated incorporation of glucose into glycogen was determined.

## Glucose utilization in isolated epididymal adipose tissue and brown adipose tissue

Distal parts of epididymal adipose tissue or interscapular brown adipose tissue were rapidly dissected and incubated for 2 hours in Krebs-Ringer bicarbonate buffer with 5 mmol/L glucose, 0.1 µCi $^{14}$C-U-glucose/mL (UVVR, Prague, Czech Republic) and 2% bovine serum albumin, gaseous phase 95% $O_2$ and 5% $CO_2$ in the presence (250 µU/mL) or absence of insulin in incubation media. All incubations were performed at 37˚C in sealed vials in a shaking water bath. Then we estimated incorporation of $^{14}$C-glucose into neutral lipids. Briefly, adipose tissue was removed from incubation medium, rinsed in saline, and immediately put into chloroform. The pieces of tissue were dissolved using a Teflon pestle homogenizer, methanol was added (chloroform:methanol 2:1), and lipids were extracted at 4˚C overnight. The remaining tissue was removed, $KH_2PO_4$ was added and a clear extract was taken for further analysis. An aliquot was evaporated, reconstituted in scintillation liquid, and the radioactivity measured by scintillation counting. Incremental glucose utilization was calculated as the difference between the insulin stimulated and basal incorporation of glucose into neutral lipids.

## Lipolysis in isolated epididymal adipose tissue

For measurement of basal and adrenaline stimulated lipolysis, the distal parts of epididymal adipose tissue were incubated in Krebs-Ringer phosphate buffer containing 3% bovine serum albumin (Armour, Fraction V) at 37˚C, pH 7.4 with or without adrenaline (0.25 µg/ml). The tissue was incubated for 2 hours and the concentrations of NEFA and glycerol in the medium were determined.

## Tissue triglyceride and cholesterol measurements

For determination of triglycerides in liver, gastrocnemius muscle, kidney, and heart, tissues were powdered under liquid $N_2$ and extracted for 16 hours in chloroform:methanol, after which 2% $KH_2PO_4$ was added and the solution was centrifuged. The organic phase was removed and evaporated under $N_2$. The resulting pellet was dissolved in isopropyl alcohol, and

triglyceride and cholesterol concentrations were determined by enzymatic assay (Pliva-Lachema, Brno, Czech Republic).

## Biochemical analyses

Blood glucose levels were measured by the glucose oxidase assay (Pliva-Lachema, Brno, Czech Republic) using tail vein blood drawn into 5% trichloracetic acid and promptly centrifuged. NEFA levels were determined using an acyl-CoA oxidase-based colorimetric kit (Roche Diagnostics GmbH, Mannheim, Germany). Serum triglyceride and cholesterol concentrations were measured by standard enzymatic methods (Pliva-Lachema, Brno, Czech Republic). Glycerol was determined using an analytical kit from Sigma. Serum insulin concentrations were determined using a rat insulin ELISA kit (Mercodia, Uppsala, Sweden).

## Parameters of oxidative stress

Activities of superoxide dismutase (SOD), glutathione peroxidase (GSH-Px), and glutathione reductase (GR) were analysed using Cayman Chemicals assay kits (MI, USA) according to manufacturer's instructions. Catalase (CAT) activity measurement was based on the ability of $H_2O_2$ to produce with ammonium molybdate a colour complex detected spectrophotometrically. The level of reduced glutathione (GSH) was determined in the reaction of SH-groups using Ellman reagent. The level of reduced (GSH) and oxidized (GSSG) form of glutathione was determined by high-performance liquid chromatography method with fluorescent detection according to HPLC diagnostic kit (Chromsystems, Munich, Germany). Lipoperoxidation products were assessed by the levels of thiobarbituric acid reactive substances (TBARS) determined by assaying the reaction with thiobarbituric acid. The levels of conjugated dienes were analysed by extraction in the media (heptane:isopropanol = 2:1) and measured spectrophotometrically in heptane's layer.

## Statistical analysis

Statistical analyses were performed in R [145]. Normality of distribution of strain means for neurogenesis phenotypes was checked using Shapiro-Wilk test. Differences between groups were tested using Student's *t*-test. Responses to insulin were analysed with a two-way linear mixed effect models using the lme4 R package [133]. Genotype, insulin treatment and interaction between genotype and treatment were used as fixed effects and individual intercepts were used as a random effect. Significance of main terms was evaluated by likelihood ratio test using function *Anova* from the car package [146], followed by Tukey post hoc test using multcomp package [147]. Values are represented as means +/- standard error of the mean. Plots were generated using the ggplot2 package [148].

## Supporting information

**S1 Fig. Metabolic phenotyping of three-month-old heterozygous SHR-*Tti2*+/- rats and wild type SHR- *Tti2*+/+ littermates (denoted as SHR).** Details of statistical analysis are in Table 2. Abbreviations: BAT, brown adipose tissue; GSH, glutathione; HDL, high-density lipoprotein; TG, triglycerides.
(TIF)

**S2 Fig. Top canonical pathways enriched among differentially expressed genes between SHR-*Tti2*+/- rats and wild type SHR littermates.** Analysis was performed with Ingenuity pathway analysis (IPA), which, in addition to calculating enrichment, predicts activation (positive Z-score, red) or inhibition (negative Z-score, blue) of molecular pathways from the

direction and magnitude of expression changes using curated database. Grey bars depict enriched pathways for which activation status could not be predicted. Vertical green dashed line indicates *p* value threshold of 0.05.
(TIF)

**S3 Fig. Predicted regulator networks effects involved in glucose homeostasis in livers of SHR-*Tti2*+/- rats.** Differentially expressed genes in livers of SHR-*Tti2*+/- rats and wild type SHR littermates were analysed using IPA. Up- (magenta) and downregulated genes (green; middle tier) connect the potential upstream regulators (upper tier) to downstream outcomes (bottom tier). Edges represent relationships derived from curated databases.
(TIF)

**S4 Fig. Frequencies of single- and multi-tissue human *TTI2* eQTL.** 821 genomic variants underlying 2177 eQTL (*p* < 1e-4) extracted from eQTL EBI catalogue (https://www.ebi.ac.uk/eqtl/) were clustered according to the number of distinct tissues or cell types in which eQTL were detected. Multiple eQTL from different data sets derived from the same cell or tissue type were scored as a single-tissue eQTL.
(TIF)

**S5 Fig. Power to detect QTL in a population of recombinant inbred strains.** Graphs depict theoretical calculations of power to detect a QTL with a minimum LOD value of 3 with various genetic effect sizes of a locus and heritability ranging from 0.2 to 0.7. Effect size is a difference in mean trait value between homozygous animals with different parental allele at this locus expressed as a proportion of the total genetic variance. Effect size of 1 implies a Mendelian trait. The calculations were performed for 25 and 30 lines with 5 biological replicates each, because several of the published traits used in our study, including serum glucose levels, were not measured in the entire family.
(TIF)

**S1 Data. Neurogenesis and metabolism in *Tti2* heterozygous KO rats.** Individual measurements of hippocampal neurogenesis and metabolic parameters in SHR-Tti2+/- and wild type SHR rats.
(CSV)

**S2 Data. Ingenuity canonical pathways.** IPA canonical pathways enriched in differentially expressed genes in SHR-*Tti2*+/- vs. wild type SHR rats in the hippocampus, liver, muscle and perirenal adipose tissue.
(XLSX)

**S3 Data. Ingenuity upstream regulators.** Affected upstream regulators predicted by IPA from differentially expressed genes in SHR-*Tti2*+/- vs. wild type SHR rats in the hippocampus, liver, muscle and perirenal adipose tissue. Activation Z-score sis deduced from the direction and magnitude of the gene expression changes.
(XLSX)

**S4 Data. Ingenuity downstream functions and diseases.** Functions enriched among differentially expressed genes in SHR-*Tti2*+/- vs. wild type SHR rats in the hippocampus, liver, muscle and perirenal adipose tissue. IPA deduces the activation Z-scores from the direction and magnitude of the gene expression changes.
(XLSX)

**S5 Data. Human *TTI2* eQTL.** Human eQTL catalogue (https://www.ebi.ac.uk/eqtl/) was queried for variants associated with changes in the *TTI2* mRNA expression. Significant eQTL

were defined by *p* value < 1e-4.
(TXT)

**S1 Table. Genes located within the confidence interval of the joint neurogenesis-serum glucose QTL on Chromosome 16. RGD ID, Rat Genome Database gene identifier.**
(DOCX)

**S2 Table. Genes with cis-eQTL within the joint neurogenesis-serum glucose QTL, and neurogenesis QTL intervals.** Expression of genes whose cis-eQTL were located within the joint neurogenesis-serum glucose QTL interval on Chromosome 16 (Chr 16: 62.1–66.3 Mb), or within neurogenesis-specific QTL interval (Chr 16: 56.8–74.3), were used for conditional mapping for these phenotypes. Although *Saraf* gene is located outside the joint phenotype QTL interval, it has a cis-eQTL within the QTL region. Conditional LOD scores and the respective difference in phenotype LOD scores for each gene are noted. Pearson's correlation coefficient values denote correlation between a phenotype and gene expression in listed tissues.
(DOCX)

**S3 Table. Quantitative RT-PCR analysis of *Tti2* mRNA expression in BN and SHR rats.** Expression values were normalised to the mean expression of *Tti2* in BN. Note that lower cycle of threshold (ΔΔCT) values indicate higher relative expression of a gene. Data is shown as means ± standard error of the mean; *t*, Student's *t*-test statistic; df, degrees of freedom.
(DOCX)

**S4 Table. Non-synonymous amino acid substitutions in the SHR *Tti2* coding sequence.** Genomic sequence of SHR rats between positions 62.1 and 66.3 Mb on chromosome 16, which cover neurogenesis-glucose QTL, was scanned for non-synonymous amino-acid substitutions compared to reference genome using Variant Visualiser in Rat Genome Database. Within this interval, missense mutations were present only in the *Tti2* gene. Conservation score ranges from 1 (highly conserved) to 0 (not conserved). SIFT score ranges from 0 (damaging) to 1 (non-damaging). Positions are according to Rnor_5.0 genome assembly.
(DOCX)

**S5 Table. RNAseq confirmed reduction of *Tti2* mRNA expression in SHR-*Tti2*[+/-] rats compared to wild type SHR littermates.**
(DOCX)

## Acknowledgments

The authors would like to thank Karel Vales, Michaela Radostna, Hana Brozka, Jan-Hendrik Claasen, Perla Leal-Galicia, Tara L. Walker, and Sara Zocher for assistance with perfusions; Anna Rumiantseva for help with histological staining; Vladimir Landa for microinjecting fertilised ova with the ZFN construct; Olena Oliyarnyk for oxidative stress analysis; Edwin Cuppen, Marieke Simonis, Kathrin Saar, and Oliver Hummel for SHR genomic variants and genotype information.

## Author Contributions

**Conceptualization:** Anna N. Senko, Hana Malínská, Lu Lu, Robert W. Williams, Gerd Kempermann.

**Data curation:** Anna N. Senko, Rupert W. Overall, Lu Lu, Robert W. Williams.

**Formal analysis:** Anna N. Senko, Rupert W. Overall, Lu Lu, Robert W. Williams.

**Funding acquisition:** Michal Pravenec, Gerd Kempermann.

**Investigation:** Anna N. Senko, Jan Silhavy, Petr Mlejnek, Hana Malínská, Martina Hüttl, Irena Marková, Klaus S. Fabel, Ales Stuchlik, Michal Pravenec.

**Methodology:** Anna N. Senko, Rupert W. Overall, Jan Silhavy, Petr Mlejnek, Klaus S. Fabel, Lu Lu, Ales Stuchlik, Robert W. Williams, Michal Pravenec.

**Project administration:** Klaus S. Fabel, Michal Pravenec, Gerd Kempermann.

**Resources:** Rupert W. Overall, Jan Silhavy, Petr Mlejnek, Hana Malínská, Martina Hüttl, Irena Marková, Ales Stuchlik, Robert W. Williams, Michal Pravenec, Gerd Kempermann.

**Software:** Anna N. Senko, Rupert W. Overall.

**Supervision:** Robert W. Williams, Gerd Kempermann.

**Validation:** Anna N. Senko.

**Visualization:** Anna N. Senko.

**Writing – original draft:** Anna N. Senko.

**Writing – review & editing:** Anna N. Senko, Rupert W. Overall, Jan Silhavy, Petr Mlejnek, Hana Malínská, Klaus S. Fabel, Lu Lu, Ales Stuchlik, Robert W. Williams, Michal Pravenec, Gerd Kempermann.

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
