## [Decision Letter · Decision Letter 0]

9 Jul 2021

Dear Dr Kempermann,

Thank you very much for submitting your Research Article entitled 'System genetics in the rat HXB/BXH family identifies Tti2 as a pleiotropic quantitative trait gene for adult hippocampal neurogenesis and serum glucose' to PLOS Genetics.

The manuscript was fully evaluated at the editorial level and by independent peer reviewers. The reviewers were appreciative of the effort involved and the quality of the manuscript, but raised concerns (some significant) that need to be addressed. As a result, we will not be able to accept this version of the manuscript, but we would be willing to review a revised version.  Your revisions would have to satisfy the reviewers before we would consider the manuscript for publication.

Should you decide to revise the manuscript for further consideration here, your revisions should address the specific points made by each reviewer.  Reviewer #1 had several major points that require attention, including one consistent with that of Reviewer #2 (degree to which use of the heterozygous ko strain addresses the nature and impact of the natural variant in Tti2; does protein made in the low expressing allele function normally).  We will require a detailed list of your responses to all review comments and a description of the changes you have made in the manuscript.  

If you decide to revise the manuscript for further consideration at PLOS Genetics, please aim to resubmit within the next 60 days, unless it will take extra time to address the concerns of the reviewers, in which case we would appreciate an expected resubmission date by email to plosgenetics@plos.org.

[LINK]

We are sorry that we cannot be more positive about your manuscript at this stage. Please do not hesitate to contact us if you have any concerns or questions.

Yours sincerely,

Wayne N. Frankel

Associate Editor

PLOS Genetics

Gregory Barsh

Editor-in-Chief

PLOS Genetics

Reviewer's Responses to Questions

**Comments to the Authors:**

Reviewer #1: SUMMARY

The authors used a recombinant inbred rat GRP (HXB/BXH; 30 strains) derived from SHR and BN rats in a systems genetic analysis of precursor cell proliferation, neuronal survival, and gene expression to map a strong candidate gene underlying hippocampal neurogenesis (dentate gyrus) and metabolic phenotypes. The QTL contained Tti2 (telo2-interacting protein 2) for which there was an eQTL and expression that correlated with both sets of traits. They made a frameshift rat KO mutant in the N-terminus of Tti2 via ZFNs on SHR background (higher Tti2 expression) and these rats showed slower CNS neurogenesis and a peripheral metabolic phenotype. Thus, they identified a gene influencing both brain plasticity and peripheral glucose metabolism.

STRENGTHS

• Efficient use of resources with this GRP (eQTLs, correlated traits in database) to unbiasedly identify correlated traits (neurogenesis, metabolic) and validate these directly in the gene mutants they generated.

• N=5-9 rats per strain for 30 or so inbred strains over 3 independent litters. 3 IP injections of BRDU (50 mg/kg), sac’ed 28 days later, histological analysis.

• Het-het breeding and Littermate controls for the KO study. 50% knockdown more closely mimics the likely human condition. Why, however, were homozygous mutants not tested? Were they not viable?

• Conditioning on gene expression (Rather than markers) is a more direct, causal assessment of the eQTL and gene in trait variation.

• Extensive analysis of Tti2 mutants at multiple levels of intermediate phenotypes that were informed by the correlations identified in the GRP – gene expression (legacy and new microarray; RNA-seq in mutants), glucose utilization, oxidative stress markers (enzyme activity), basal and insulin-stimulated glycogen synthesis, biochemical (glucose, triglycerides, cholesterol, glycerol, insulin)

• Dose-dependent effects of Tti2 on HC neurogenesis and glucose homeostasis

• Striking degree of co-mapping of metabolic, neurogenesis, and Tti2 expression strengthens the case for Tti2.

• Importantly, proliferation did not map to chr. 16, indicating some degree of specificity for neurogenesis.

• Neat that conditioning on glucose levels decreased the LOD score below significance. Strengthens the argument that a shared genetic factor in chr. 16 QTL mediates both phenotypes. Although not slam-dunk evidence (even if they’re separate QTLs, there will be a high degree of local LD), it is complementary evidence. Are there any other transcripts with cis-eQTLs within the chr.16 region? Have you tried conditioning on those?

MAJOR COMMENTS

• The word pleiotropy is used to describe the findings but is it possible that the metabolic phenotype leads to changes in neurogenesis; in this case it would be pleiotropy or would it be more of a linear causal chain of events. Do the authors have any insight as to whether there is independent, concomitant regulation of both traits (pleiotropy) or whether is a causal sequence?

• Is Tti2 the only eQTL within the 4 Mb interval? If not, the other cis-eQTLs could be provided for all tested tissues, along with the effect of conditioning on expression of those genes on the chr.16 QTL.

• The 11 protein-coding genes and 4 noncoding RNA genes within the QTL interval should be listed somewhere as well.

• Interesting that neg. correl was observed for Tti2 exp vs. neurogenesis in the GRP, yet decreased exp in the Tti2 mutants resulted in decreased neurogenesis. Could there be a dominant negative effect with one or more of the coding polymorphisms that explains this discrepancy between the GRP and the mutants?

• If the only missense mutations within this entire 4+Mb interval are within one gene – Tti2, then I don’t think this GRP is as representative of human genetic diversity as the authors claim in the Introduction. How many polymorphisms segregate in this GRP?

• Testing males-only is a major limitation of the study as there are known sex differences in neurogenesis. “Compatibility” with prior data and “estrus variation” are not valid scientific reasons to exclude half the population. Covariates can be included in the statistical models if necessary. If not females now for these reasons, then when?

• Line 254: “…presumably generating a non-functional protein”. That is a big presumption. Is there a decrease in TTI2 protein in the mutants? Is there a decrease (or potentially increase?) in TTI2 protein in BN vs. SHR?

MINOR

• QTL interval should be provided in the abstract, if possible.

• Power analysis for QTL detection of various effect sizes with 30 strains could be provided.

• Line 228: “expression of only one gene…” – do they mean one gene within the chr.16 QTL interval? Or only one gene in the entire genome?

• Line 698: PMID listed. needs to have the reference inserted.

• Figure legends could be located closer to the figures.

Reviewer #2 SUMMARY

This is an interesting and thorough manuscript that uses rat RI strains to map a trait locus for adult neurogenesis and identify a strong candidate gene by testing a rat knockout strain generated for this purpose.  The primary genetic association is not the strongest, and this particular RI panel has only modest detection power and resolution, but the authors brought sophisticated conditional and expression QTL mapping into the mix to obtain convincing evidence that a candidate gene – Tti2 - is the likely culprit, and then introduced several levels of characterization to examine the relationship between neurogenesis and serum glucose.

The “proof” that Tti2 gene is the causal gene is almost, but not quite ironclad – more ideal would have been to knockin the most likely (or multiple) missense variants rather than to use the knockout, because of the possibility that loss-of-function is not the sole molecular consequence of the strain variation in Tti2.  I do understand, on balance, since there is a clear expression difference in Tti2 between the progenitor strains, that using the ko to test it is reasonable. However, if the authors have or would make compound heterozygotes between the knockout allele and representative RI lines for each progenitor allele (an approach previously referred to as RIST, recombinant inbred segregation testing) this would further strengthen their conclusions and the manuscript.

One minor point – the authors adoption of the p < 0.63 guideline as “suggestive” bothers me. Although I realize it has historical context from earlier QTL rigor standard-seeking guidelines, outside of the QTL world I do not think anyone in their right mind would think that this is suggestive or trending.  Depicting p<0.1 and/or p<0.25 levels seem more realistic ways to report less-than-statistically-significant effects.

**Have all data underlying the figures and results presented in the manuscript been provided?**

Reviewer #1: Yes

Reviewer #2: Yes

PLOS authors have the option to publish the peer review history of their article (what does this mean?). If published, this will include your full peer review and any attached files.

Reviewer #1: **Yes: **Camron D. Bryant

Reviewer #2: **No**.

---

## [Decision Letter · Decision Letter 1]

7 Mar 2022

Dear Dr Kempermann,

We are pleased to inform you that your manuscript entitled "System genetics in the rat HXB/BXH family identifies Tti2 as a pleiotropic quantitative trait gene for adult hippocampal neurogenesis and serum glucose" has been editorially accepted for publication in PLOS Genetics. Congratulations!

Before your submission can be formally accepted and sent to production you will need to complete our formatting changes, which you will receive in a follow up email.

Also please do consider Reviewer I's comment regarding use of male-only studies, and if there is something you would like to alter in the text accordingly, you are welcome to do so.

Please be aware that it may take several days for you to receive this email; during this time no action is required by you. Please note: the accept date on your published article will reflect the date of this provisional acceptance, but your manuscript will not be scheduled for publication until the required changes have been made.

Yours sincerely,

Wayne N. Frankel

Associate Editor

PLOS Genetics

Gregory Barsh

Editor-in-Chief

PLOS Genetics

Comments from the reviewers (if applicable):

Reviewer's Responses to Questions

**Comments to the Authors:**

Reviewer #1: Appreciatively, the authors thoroughly responded to my critiques, with one notable exception. The reasons provided for excluding females are not valid scientific reasons. 1) it's too much work; 2) it's too much variance; 2) we wouldn't be able to compare females to historical male-only data; 4) it's a GRP, we can do it later. I encourage the authors to check out DOI: 10.1126/science.aaw7570 and key references within. I chose "accept" but note that the field has rapidly and rightly moved on from male-only studies.

**Have all data underlying the figures and results presented in the manuscript been provided?**

Reviewer #1: Yes

PLOS authors have the option to publish the peer review history of their article (what does this mean?). If published, this will include your full peer review and any attached files.

Reviewer #1: **Yes: **Camron D Bryant

**Data Deposition**

http://datadryad.org/submit?journalID=pgenetics&manu=PGENETICS-D-21-00750R1

**Press Queries**

---

## [Editor Report · Acceptance letter]

30 Mar 2022

PGENETICS-D-21-00750R1 

Systems genetics in the rat HXB/BXH family identifies Tti2 as a pleiotropic quantitative trait gene for adult hippocampal neurogenesis and serum glucose 

Dear Dr Kempermann, 

We are pleased to inform you that your manuscript entitled "Systems genetics in the rat HXB/BXH family identifies Tti2 as a pleiotropic quantitative trait gene for adult hippocampal neurogenesis and serum glucose" has been formally accepted for publication in PLOS Genetics! Your manuscript is now with our production department and you will be notified of the publication date in due course.

With kind regards,

Anita Estes

PLOS Genetics

On behalf of:
